# TNF-α-mediated m⁶A modification of ELMO1 triggers directional migration of mesenchymal stem cell in ankylosing spondylitis

Zhongyu Xie[1,3], Wenhui Yu[1,3], Guan Zheng[1,3], Jinteng Li[1], Shuizhong Cen[1], Guiwen Ye[1], Zhaofeng Li[1], Wenjie Liu[1], Ming Li[1], Jiajie Lin[1], Zepeng Su[1], Yunshu Che[1], Feng Ye[1], Peng Wang[1✉], Yanfeng Wu[2✉] & Huiyong Shen [1✉]

Ankylosing spondylitis (AS) is a type of rheumatic disease characterized by chronic inflammation and pathological osteogenesis in the entheses. Previously, we demonstrated that enhanced osteogenic differentiation of MSC from AS patients (AS-MSC) resulted in pathological osteogenesis, and that during the enhanced osteogenic differentiation course, AS-MSC induced TNF-α-mediated local inflammation. However, whether TNF-α in turn affects AS-MSC remains unknown. Herein, we further demonstrate that a high-concentration TNF-α treatment triggers enhanced directional migration of AS-MSC in vitro and in vivo, which enforces AS pathogenesis. Mechanistically, TNF-α leads to increased expression of ELMO1 in AS-MSC, which is mediated by a METTL14 dependent m⁶A modification in *ELMO1* 3'UTR. Higher ELMO1 expression of AS-MSC is found in vivo in AS patients, and inhibiting ELMO1 in SKG mice produces therapeutic effects in this spondyloarthritis model. This study may provide insight into not only the pathogenesis but also clinical therapy for AS.

[1] Department of Orthopedics, The Eighth Affiliated Hospital, Sun Yat-sen University, 3025# Shennan Road, Shenzhen 518000, P.R. China. [2] Center for Biotherapy, The Eighth Affiliated Hospital, Sun Yat-sen University, 3025# Shennan Road, Shenzhen 518000, P.R. China. [3] These authors contributed equally: Zhongyu Xie, Wenhui Yu, Guan Zheng. ✉email: wangp57@mail.sysu.edu.cn; wuyf@mail.sysu.edu.cn; shenhuiy@mail.sysu.edu.cn

Ankylosing spondylitis (AS) is a common kind of rheumatic disease that affects 0.1–0.5% of the population worldwide[1]. As the major component of spondyloarthropathy, AS is characterized by chronic inflammation and ectopic ossification in the entheses, which distinguishes AS from other rheumatic diseases[2]. However, both the mechanisms of chronic inflammation and ectopic ossification and the relationship between these two features still require further investigation.

Mesenchymal stem cells (MSC) are a kind of pluripotent adult stem cell found in vivo[3]. As both a major source of osteoblasts and a critical regulator of immune cells, MSC play an important role as a bridge connecting bone metabolism and immune homeostasis[4–6]. Previously, our study, as well as Chin-Hsiu Liu's results, indicated that the abnormally enhanced osteogenic differentiation capacity of MSC from AS patients (AS-MSC) resulted in pathological osteogenesis and syndesmophyte formation[7,8]. Moreover, we further demonstrated that AS-MSC secreted a relatively large amount of CCL2 during the enhanced osteogenic differentiation course, which augmented monocyte migration, increased proinflammatory macrophage polarization and enhanced TNF-α secretion at the site of enthesis[9]. However, whether TNF-α secreted by macrophages, in turn, affects AS-MSC is still an open question.

Directional migration to a specific location ensures that MSC exert powerful functions[10]. Since normal MSC migration contributes to tissue regeneration and immune regulation, it is unsurprising that directional migration of dysfunctional MSC results in disease development. The migration process is under the regulation of various factors[11]. Engulfment and cell motility protein 1 (ELMO1), which is known as CED12 in *Caenorhabditis elegans*, is a core molecule involved in cell migration[12]. ELMO1 functionally binds dedicator of cytokinesis (DOCK) proteins and then regulates cytoskeletal rearrangement via Rac1 activation, which participates in various biological functions, such as directional migration, engulfment of apoptotic cells, nervous system development and cancer cell invasion[12–15]. Previous studies have shown that the abnormal expression of ELMO1 is greatly involved in several diseases, including diabetic nephropathy, inflammatory arthritis and inflammatory bowel disease[16–18]. Both the role of ELMO1 in MSC directional migration and its effect on AS pathogenesis are still ambiguous.

$N^6$-methyladenosine (m6A), present at approximately three to five sites per mRNA transcript (0.1–0.4% of adenosines) in mammalian cells, is the most abundant internal mRNA modification[19]. m6A modification regulates gene expression and is widely involved in cell function and development[20,21]. Differential expression of methyltransferases such as METTL3 and METTL14 or demethylases including FTO and ALKBH5 results in abnormal expression of specific genes, which ultimately contributes to diseases development[22–24]. However, the effect of m6A modification on AS pathogenesis has never been investigated.

In this study, we demonstrate that a high level of TNF-α induced stronger directional migration of AS-MSC compared to MSC from healthy controls (HC-MSC) by increasing ELMO1 expression. Further research showed that this phenomenon resulted from the METTL14 mediated m6A modification of the *ELMO1* 3′UTR in AS-MSC after TNF-α treatment. A higher expression of ELMO1 in AS-MSC was also found in the enthesis of AS patients in vivo. Moreover, injecting Av-ELMO1 significantly improved inflammation and ectopic ossification in SKG mice serving as a spondyloarthritis model. This study may not only contribute to the elucidation of the pathogenesis of AS but also provide insight into clinical therapy for AS.

## Results

### TNF-α induces enhanced directional migration of AS-MSC in vitro.
AS-MSC were isolated from 15 patients during the active period of AS, and HC-MSC were obtained from 15 healthy donors matched by age and gender (Supplementary Table 1). Previously, we demonstrated that AS-MSC augmented monocyte migration and increased M1 proinflammatory macrophage polarization[9]. In this study, the migration assays showed that the numbers of stained migratory AS-MSC were much greater than those of HC-MSC when cocultured with macrophages (Fig. 1a). This discrepancy was rectified by a TNF-α neutralizing antibody, but not an anti-IL17 or anti-IL23 neutralizing antibody, with equal levels observed in AS-MSC and HC-MSC (Fig. 1a and Supplementary Fig. 1), indicating that macrophage-secreted TNF-α affected AS-MSC by enhancing their migratory capacity. After treatment with 100 ng/ml TNF-α, the number of stained migratory AS-MSC was much greater than that of HC-MSC. No difference between HC-MSC and AS-MSC was observed with or without TNF-α stimulation at lower concentrations (0, 10, and 50 ng/ml) (Fig. 1b). Consistently, as shown by a wound-healing assay, AS-MSC treated with 100 ng/ml TNF-α had a larger migratory area percentage than HC-MSC (Fig. 1c). To further investigate the directional migration ability, a μ-slide chemotaxis assay was performed. Cell tracking showed that AS-MSC were more likely to migrate toward TNF-α than were HC-MSC. The migration index, velocity, and directionality were all higher for AS-MSC than for HC-MSC (Fig. 1d). These results suggest that TNF-α at a relatively high concentration led to enhanced directional migration of AS-MSC in vitro.

### TNF-α accelerates the directional migration of AS-MSC in vivo.
To further investigate TNF-α-induced AS-MSC migration in vivo, we transplanted MSC/Fluc and injected TNF-α into nude mice, followed by the performance of a bioluminescence assay and an immunohistochemical assay (Fig. 2a). The MSC migration area shown by the bioluminescence assay gradually increased from day 0 to day 5. On days 3 and 5 after MSC injection, the bioluminescence area of AS-MSC was larger than that of HC-MSC (Fig. 2b). Moreover, as shown by the immunohistochemical assay, the number of migratory AS-MSC, identified with an anti-human HLA antibody, was much greater than that of migratory HC-MSC at the TNF-α injection sites (Fig. 2c). These results indicated that AS-MSC had a stronger migratory capacity than HC-MSC with high-concentration TNF-α treatment.

### TNF-α leads to elevated expression of ELMO1 in AS-MSC.
The whole transcriptome sequencing of six HC-MSC and six AS-MSC, without or with 100 ng/ml TNF-α stimulation, was performed. A total of 2631 differentially expressed mRNAs were identified in the HC-MSC, and 4680 differentially expressed mRNAs were identified in the AS-MSC before and after TNF-α stimulation. When comparing HC-MSC to AS-MSC after 100 ng/ml TNF-α stimulation, a total of 1780 differentially expressed mRNAs were found (Supplementary Fig. 2A). Clustergram analysis showed that significant differences were observed between the MSC treated without and with TNF-α and between the AS-MSC and HC-MSC treated with TNF-α (Fig. 3a). Consistently, PCA confirmed that AS-MSC treated with 100 ng/ml TNF-α had a different expression profile than HC-MSC (Fig. 3b).

The mRNAs that were differentially expressed between both HC-MSC (or AS-MSC) with and without TNF-α and between TNF-α-treated AS-MSC and HC-MSC were considered to be critical genes. Through Venn diagram analysis, 489 critical differentially expressed mRNAs were identified (Fig. 3c). As determined by GO analysis, these 489 mRNAs were enriched in the inflammatory response, chemotaxis and the chemokine-mediated signaling pathway in the biological process category and

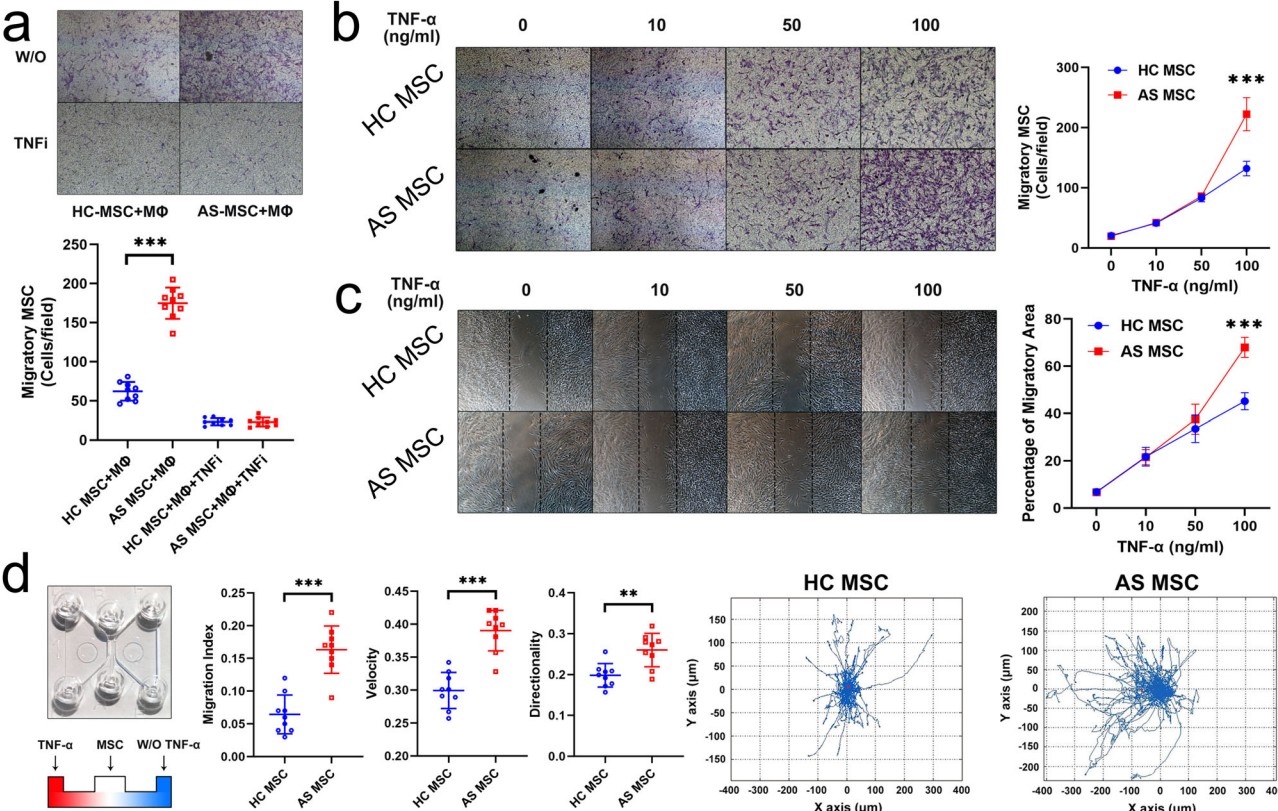

**Fig. 1 TNF-α induces enhanced directional migration of AS-MSC in vitro. a** The number of stained migratory AS-MSC ($n = 9$) was larger than that of stained migratory HC-MSC ($n = 9$) when the MSC were cultured with macrophages (Mø; $P = 1.2793E-10$). The number of stained migratory HC-MSC with an anti-TNF-α neutralizing antibody (TNFi) ($n = 9$) were equal to that of stained migratory AS-MSC with TNFi ($n = 9$; $P = 0.892$). **b** The number of stained migratory AS-MSC ($n = 9$) was larger than that of stained migratory HC-MSC ($n = 9$) when the MSC were treated with 100 ng/ml TNF-α ($P = 1.1836E-7$). **c** The migratory area of MSC increased after TNF-α stimulation, and the migratory area of AS-MSC ($n = 9$) was larger than that of HC-MSC ($n = 9$) upon treatment with 100 ng/ml TNF-α ($P = 1.4639E-9$). **d** In a μ-slide chamber, the migration index ($P = 0.00001$), velocity ($P = 0.002$) and directionality ($P = 0.000006$) of AS-MSC ($n = 9$) toward TNF-α were all higher than those of HC-MSC ($n = 9$). The motion curve of AS-MSC toward TNF-α showed a stronger tendency than that of HC-MSC. Data were analyzed using two-tailed Student's $t$-test. Values are presented as the mean ± SD. ** Indicates $P < 0.01$, and *** indicates $P < 0.001$. W/O TNF-α means without TNF-α.

enriched in chemokine activity and cytokine activity in the molecular function category (Supplementary Fig. 2B). Moreover, "cytokine–cytokine receptor interaction", "chemokine signaling pathway", and "cell adhesion molecular" were the top 3 enriched pathways by KEGG analysis (Fig. 3d). Deep analysis of these pathways identified 38 differentially expressed mRNAs, among which ELMO1 had the largest fold change (4.16-fold upregulated in AS-MSC compared to HC-MSC after TNF-α treatment) and was the central intersection point of the chemokine signaling pathway (Fig. 3e and Supplementary Fig. 2C).

Consistent with the sequencing data, PCR results showed that ELMO1 expression in MSC increased after TNF-α treatment and that its expression in TNF-α-treated AS-MSC was much higher than that in TNF-α-treated HC-MSC (Fig. 3f). These results were also confirmed at the protein level by western blotting (Fig. 3g). The cell immunofluorescence assay revealed that the fluorescence intensity of ELMO1 after the TNF-α stimulation in AS-MSC was higher than that in HC-MSC (Fig. 3h). Regarding other ELMOs and their binding protein DOCKs, ELMO3 and DOCK8 were not expressed in MSC, and no differences in ELMO2 or DOCK1, 2, 4, and 5 were observed between HC-MSC and AS-MSC (Supplementary Fig. 3).

Studies have demonstrated that ELMO1 activates Rac1 and promotes chemotaxis by binding to DOCK proteins in other cells[15,17]. The active Rac1 level in MSC was also enhanced with increasing TNF-α concentrations. Notably, the active Rac1 level

in AS-MSC was significantly higher than that in HC-MSC treated with 100 ng/ml TNF-α (Fig. 3i). In addition, the protein-protein interactions identified using a Co-IP and LC-MS/MS assay showed that ELMO1 mainly bound DOCK1, DOCK4, DOCK5, ELMO2, and NCKAP1 in MSC, which is similar to the interactions previously reported in other cells, and these interaction were the same in HC-MSC and AS-MSC (Supplementary Fig. 4A–C and Supplementary Tables 5 and 6). Moreover, inhibiting DOCK1, but not DOCK4 and 5, counteracted the active Rac1 level upregulated by ELMO1 overexpression, indicating that ELMO1 bound DOCK1 to promote Rac1 activation in the TNF-α treated MSC (Supplementary Fig. 4D).

**TNF-α mediated ELMO1 expression promotes MSC migration.** Then, we investigated the role of ELMO1 in MSC migration with TNF-α treatment. Three siRNAs specific for ELMO1 were designed, and siRNA2, which had the highest efficiency, was used to construct Lv-ELMO1; the inhibitory efficiency of Lv-ELMO1 was confirmed at the gene and protein levels (Supplementary Fig. 6A–C). No difference in proliferation ability was observed between the NC and Lv-ELMO1 group (Supplementary Fig. 6D). After inhibiting ELMO1 in AS-MSC treated with 100 ng/ml TNF-α, the number of stained migratory cells in the transwell assay and the migratory area in the wound-healing assay were equal to the

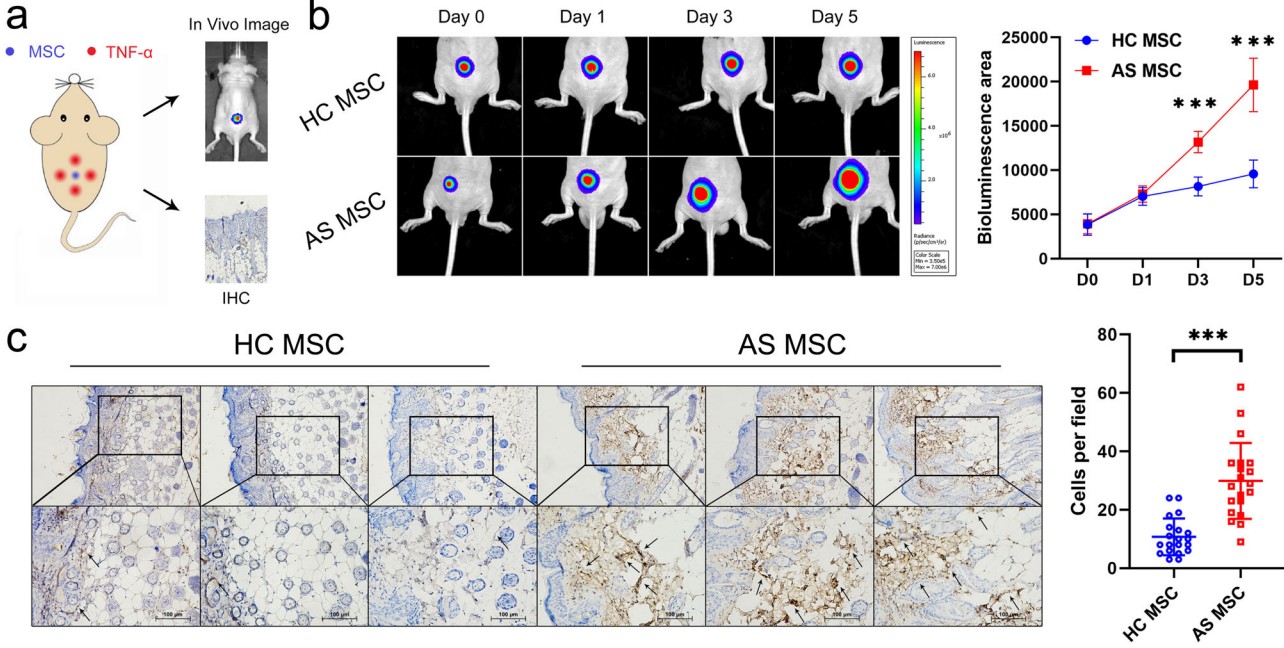

**Fig. 2 TNF-α accelerates the directional migration of AS-MSC in vivo. a** Schematic diagram of the in vivo migration assay. MSC were transplanted into nude mice, and TNF-α was injected around the MSC. In vivo imaging and an immunohistochemical (IHC) assay were then performed. **b** The bioluminescence area of MSC increased from day 0 to day 5, and the bioluminescence area of AS-MSC ($n = 9$) was larger than that of HC-MSC ($n = 9$) on days 3 ($P = 6.6784E-8$) and 5 ($P = 1.9895E-14$) after injection. **c** The number of migratory AS-MSC determined by counting cells per field ($n = 20$) was higher than that of migratory HC-MSC ($P = 7.1465E-7$). Data were analyzed using two-tailed Student's $t$-test. Values are presented as the mean ± SD. *** Indicates $P < 0.001$. Scar bar = 100 µm.

results for HC-MSC (Fig. 4a, b). In addition, the bioluminescence area of Lv-ELMO1-transfected AS-MSC in nude mice was also reduced compared to that of Lv-NC-transfected AS-MSC and showed no difference from that of HC-MSC (Fig. 4c). The downstream active Rac1 level in Lv-ELMO1-transfected AS-MSC treated with TNF-α was also markedly weakened to a level equal to that of HC-MSC treated with TNF-α (Fig. 4d).

An OE-ELMO1 vector was constructed, and the overexpression efficiency was also confirmed at the gene and protein levels (Supplementary Fig. 6E, F). The proliferative abilities of the NC group and the OE-ELMO1 group were equal (Supplementary Fig. 6G). When ELMO1 expression was increased in HC-MSC, their migratory ability, as shown by transwell and wound-healing assays, was observably improved with 100 ng/ml TNF-α stimulation and almost reached the ability of AS-MSC (Fig. 4e, f). These results were further confirmed by evaluating the bioluminescence migratory area (Fig. 4g). Similar to AS-MSC, OE-ELMO1-transfected HC-MSC had higher active Rac1 levels than control HC-MSC (Fig. 4h). These results suggest that ELMO1 promotes MSC migration and that abnormal ELMO1 upregulation mediated by TNF-α stimulation results in enhanced AS-MSC migration.

**ELMO1 acts as a therapeutic target in AS.** Enthesis were obtained from nine AS patients and 9 matched non-AS patients during lumbar surgery (Supplementary Table 4). ELMO1 expression (as shown by an immunofluorescence assay and quantified by the mean fluorescence intensity) in CD105[+] MSC in the AS enthesis samples was higher than that in the control enthesis samples (Fig. 5a). Then, SKG mice were treated with 3 mg curdlan via an intraperitoneal injection to induce the AS feature, and Av-ELMO1 was constructed for their treatment. After injection of Av-ELMO1, the arthritis score were significantly reduced, and the disease incidence time was delayed in

the injected group compared to the control groups (Fig. 5b, c). Moreover, local inflammation which was shown by monitoring clinical manifestations and performing H&E staining, and ectopic ossification which was shown by micro-CT examination, were also improved by Av-ELMO1 (Fig. 5d–f). When focusing on the local tissue of ankle enthesis, the numbers of migratory CD105[+] MSC were markedly decreased in the Av-ELMO1 group. In addition, the numbers of CD68[+] macrophages and expression of TNF-α were also reduced by Av-ELMO1 in SKG mice (Fig. 5g).

**Decreasing METTL14 expression leads to lower m[6]A modification and RNA degradation of *ELMO1* in TNF-α-treated AS-MSC.** Previous study determined that mRNA stability and expression can be negatively regulated by m[6]A modification[25]. Herein, we found that the m[6]A modification level of *ELMO1* in MSC was reduced with increasing TNF-α concentrations and that after 100 ng/ml TNF-α treatment, the *ELMO1* m[6]A modification level in AS-MSC was much lower than that in HC-MSC (Fig. 6a). In addition, the *ELMO1* mRNA half-life of AS-MSC was longer than that of HC-MSC with 100 ng/ml TNF-α stimulation, indicating better mRNA stability in the AS-MSC (Fig. 6b, c). Then, we detected the expression levels of m[6]A-related methyltransferase and demethylases. The results showed that METTL3, ALKBH5, and FTO expression increased and METTL14 and WTAP expression decreased after TNF-α treatment. When treated with 100 ng/ml TNF-α, AS-MSC had lower METTL14 expression than HC-MSC. No significant differences were observed in other m[6]A-related methyltransferases and demethylases (Fig. 6d, e, Supplementary Fig. 5). Consistently, METTL14 expression in CD105[+] MSC in the AS enthesis samples was significantly lower than that in the control enthesis samples (Fig. 6f). These results indicated that the lower METTL14 expression in TNF-α-treated AS-MSC than in HC-MSC may lead to a lower m[6]A level and slower degeneration rate of *ELMO1*.

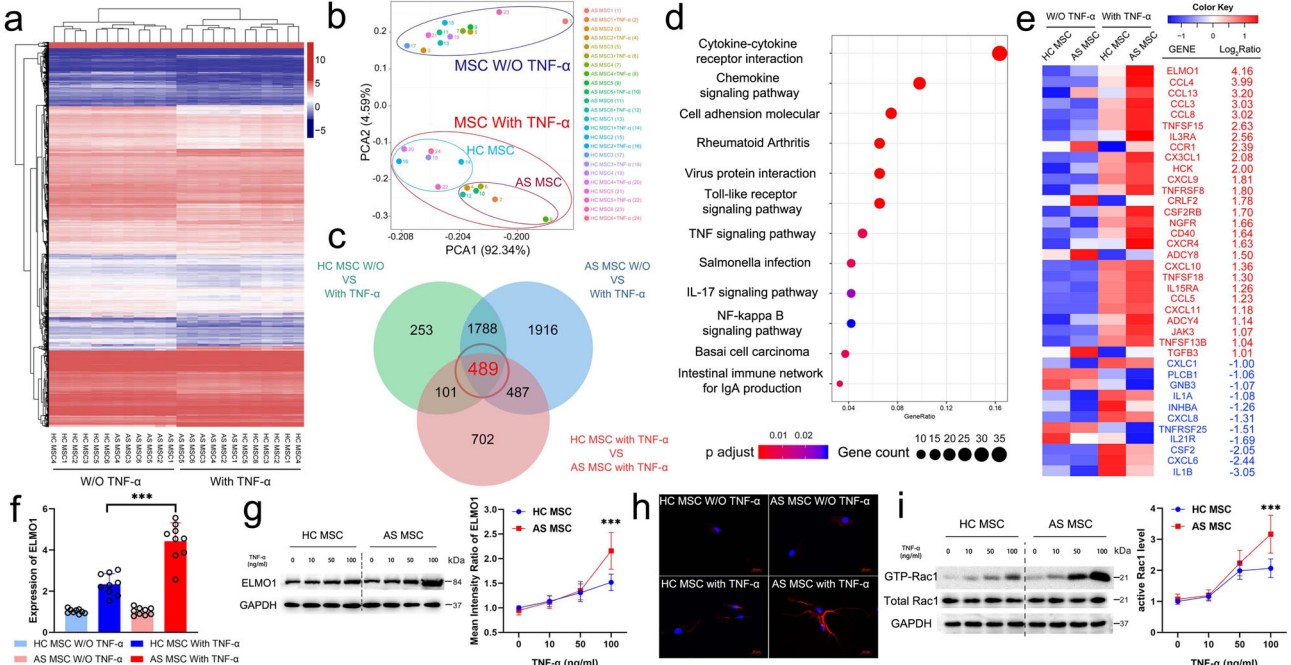

**Fig. 3 TNF-α leads to elevated expression of ELMO1 in AS-MSC. a** Cluster heatmap of HC-MSC ($n = 6$) and AS-MSC ($n = 6$) without and with 100 ng/ml TNF-α treatment. **b** PCA analysis of HC-MSC ($n = 6$) and AS-MSC ($n = 6$) without (W/O) and with 100 ng/ml TNF-α treatment. **c** Venn diagram of HC-MSC or AS-MSC without (W/O) and with 100 ng/ml TNF-α treatment. **d** The top 10 significant pathways enriched from the 489 critical genes in KEGG analysis. **e** Heatmap and fold changes in the 27 upregulated genes and 11 downregulated genes in the chemokine signaling pathway. **f** ELMO1 mRNA expression in AS-MSC ($n = 9$) was higher than that in HC-MSC ($n = 9$) after 100 ng/ml TNF-α treatment. $P = 1.075E-8$ based on one-way ANOVA followed by Bonferroni's post hoc comparisons. **g** ELMO1 expression in AS-MSC ($n = 9$) was higher than that in HC-MSC ($n = 9$) after 100 ng/ml TNF-α treatment at the protein level. $P = 0.000253$ based on two-tailed Student's $t$-test. **h** Immunofluorescence results showing that the fluorescence intensity of ELMO1 was stronger in AS-MSC than in HC-MSC after 100 ng/ml TNF-α treatment. Red fluorescence indicated ELMO1 expression and blue fluorescence indicated cell nucleus. Experiment was repeated nine time independently. Scar bar = 20 μm. **i** After treatment with 100 ng/ml TNF-α, the active Rac1 level of AS-MSC ($n = 9$) was higher than that of HC-MSC ($n = 9$). $P = 0.00018$ based on two-tailed Student's $t$-test. Values are presented as the mean ± SD. *** Indicates $P < 0.001$. W/O TNF-α means without TNF-α.

**Inhibiting METTL14 expression increases ELMO1 expression and promotes MSC directional migration.** The most efficient siRNA specific for METTL14 was chosen to construct Lv-M14 (Supplementary Fig. 6H). The inhibitory efficiency of Lv-M14 was confirmed at the protein level, and inhibiting METTL14 in MSC significantly increased ELMO1 expression but did not affect MSC proliferation (Fig. 7a, Supplementary Fig. 6I). In addition, the results of m6A RIP PCR showed that the m6A modification level of *ELMO1* was markedly lower in Lv-M14-transfected MSC than in Lv-NC-transfected MSC (Fig. 7B). Moreover, inhibiting METTL14 expression in MSC prolonged the *ELMO1* mRNA half-life compared to control treatment (Fig. 7c, d). The active Rac1 level in MSC was also upregulated after Lv-M14 transfection (Fig. 7e). Then, we investigated whether inhibiting METTL14 affects the migratory ability of MSC. As expected, the number of stained migratory MSC and the migratory area in a wound-healing assay were much higher in the Lv-M14 group than in the Lv-NC group (Fig. 7f, g). In addition, the bioluminescence area of Lv-M14-transfected MSC was also increased in nude mice (Fig. 7h).

**METTL14 overexpression downregulates ELMO1 levels and inhibits MSC directional migration.** To further study the roles of METTL14 in ELMO1 expression and MSC migration, OE-M14 was used, and its efficiency and effect on MSC proliferation was verified (Supplementary Fig. 4J, K). Increasing METTL14 expression obviously inhibited ELMO1 expression in MSC (Fig. 8a). The m6A modification of *ELMO1* in METTL14-

overexpressing MSC was increased (Fig. 8b). Additionally, the *ELMO1* mRNA half-life in MSC was shortened after OE-M14 transfection (Fig. 8c, d). The active level of Rac1 in OE-M14-transfected MSC was reduced compared to that in control MSC (Fig. 8e). When METTL14 was overexpressed, the migratory abilities of MSC in vitro, as shown by the Transwell migration and wound-healing assays, and in vivo, as shown by the bioluminescence area measurement, were inhibited (Fig. 8F–H). The results shown in Figs. 7 and 8 suggest that METTL14 enhances *ELMO1* m6A modification and accelerates *ELMO1* mRNA degeneration, which then decreases ELMO1 expression to inhibit MSC migration.

**METTL14 acts on the specific m6A modification site in the *ELMO1* 3′UTR.** We further investigated the mechanism by which METTL14 accelerates the degeneration rate of *ELMO1* mRNA. The CLIP-PCR results showed that *ELMO1* was more enriched in the METTL14 group than the IgG group, suggesting that METTL14 could specifically bind to the *ELMO1* mRNA transcript in MSC (Fig. 9a). A previous study indicated that METTL14 mainly bound to the 3′UTR of mRNAs and promoted mRNA degeneration[26]. In a dual-luciferase reporter assay, the luciferase activity of the reporter plasmid with the *ELMO1* 3′UTR was significantly reduced by wild-type but not mutant METTL14 compared to that of the control group (Fig. 9b). In addition, a reporter plasmid with a mutated *ELMO1* 3′UTR was constructed to elucidate the m6A modification site of *ELMO1* (Fig. 9c). The luciferase activity of only the reporter plasmid with mutant 1

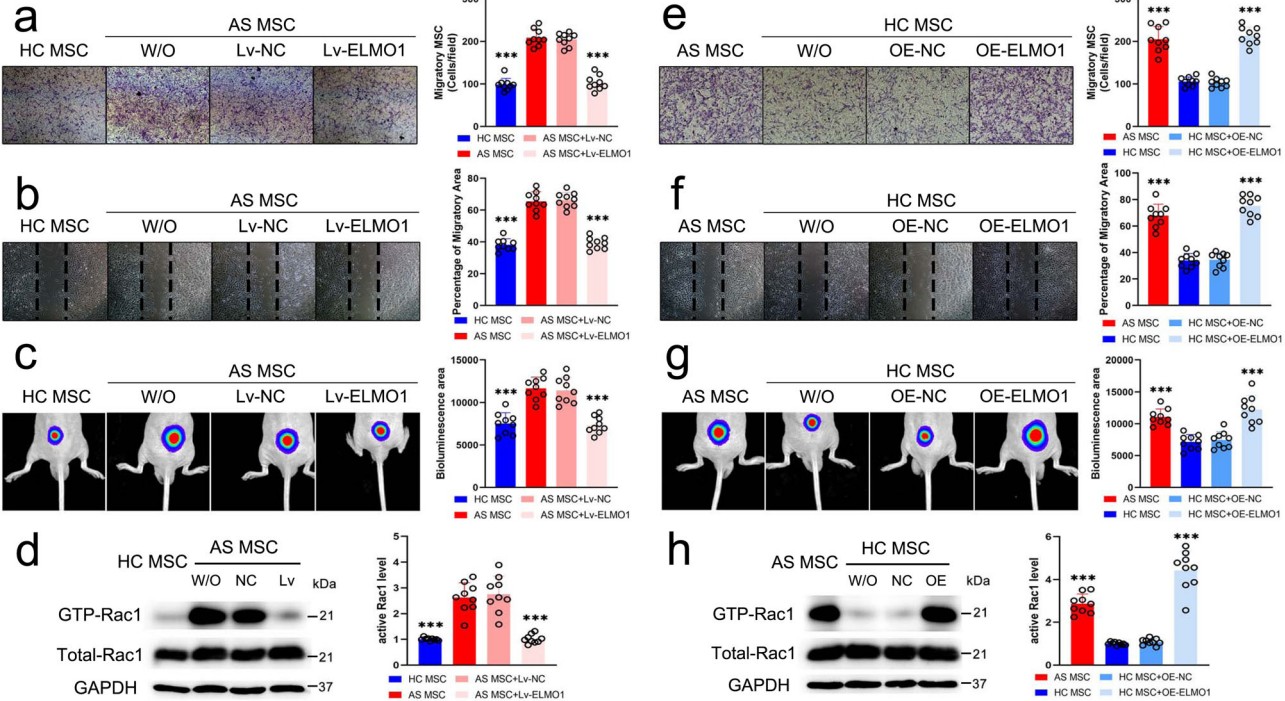

**Fig. 4 TNF-α-mediated ELMO1 expression promotes MSC migration. a** The number of stained migratory AS-MSC ($n = 9$) in Lv-ELMO1 group was lower than Lv-NC group ($n = 9$; $P = 2.677E{-}13$). **b** The migratory area of AS-MSC in Lv-ELMO1 group ($n = 9$) was decreased compared to the Lv-NC group ($n = 9$; $P = 5.0514E{-}13$). **c** The bioluminescence area of AS-MSC ($n = 9$) in Lv-ELMO1 group was significantly reduced compared to the Lv-NC group ($n = 9$; $P = 8.3309E{-}7$). **d** The active Rac1 level of AS-MSC in Lv-ELMO1 group ($n = 9$) was decreased compared to the Lv-NC group ($n = 9$; $P = 2.9435E{-}8$). **e** The number of stained migratory HC-MSC ($n = 9$) in OE-ELMO1 group was higher than the OE-NC group ($n = 9$; $P = 2.7053E{-}12$). **f** The migratory area of HC-MSC in OE-ELMO1 group ($n = 9$) was also higher than the OE-NC group ($n = 9$; $P = 5.7117E{-}13$). **g** The bioluminescence area of OE-ELMO1 HC-MSC group ($n = 9$) was significantly elevated compared to the OE-NC group ($n = 9$; $P = 0.000001$). **h** The active Rac1 level of OE-ELMO1-transfected HC-MSC was enhanced compared to the OE-NC group ($n = 9$; $P = 1.6488E{-}9$). Data were analyzed using on one-way ANOVA followed by Bonferroni's post hoc comparisons. Values are presented as the mean ± SD. *** Indicates $P < 0.001$. W/O indicates MSC without lentivirus transfection. Lv-NC or OE-NC indicates MSC transfected with control lentiviruses. Lv-ELMO1 indicates MSC transfected with lentiviruses encoding an shRNA specific for ELMO1. OE-ELMO1 indicates MSC transfected with lentiviruses overexpressing ELMO1.

*ELMO1* 3′UTR was restored to the normal level seen in the control group. The luciferase activities of the reporter plasmids with mutant 2 and 3 *ELMO1* 3′UTR were equal to the activity of the reporter plasmid with wild-type *ELMO1* 3′UTR (Fig. 9d). Through CLIP assays, *ELMO1* mRNA was found to bind YTHDF2 and YTHDF3 but not YTHDF1, YTHDC1 or YTHDC2 (Supplementary Fig. 7A). Additionally, inhibiting YTHDF2 or YTHDF3 decreased the *ELMO1* mRNA degeneration rate, indicating that METTL14 regulates *ELMO1* mRNA through the m$^6$A reader enzymes YTHDF2 and YTHDF3 (Supplementary Fig. 7B).

## Discussion

AS is a common kind of rheumatic disease with a high prevalence rate of 0.1–0.5% as well as high costs of illness[1,27]. Distinct from other rheumatic diseases, AS has the hallmark clinical feature called enthesitis, which is the pathological basis of chronic inflammation and ectopic ossification[2]. However, both the pathogenesis of enthesitis and the relationship between chronic inflammation and ectopic ossification are still unclear. As both the major source of osteoblasts and the critical regulator of immune cells, MSC play an important role as the bridge between bone metabolism and immune homeostasis[4,6]. Previous studies have demonstrated that dysfunctions in MSC contribute to the development of rheumatic diseases[28,29]. Our work and studies by other teams have determined that AS-MSC possess an enhanced osteogenic differentiation capacity, which is one of the critical pathogenic processes of ectopic

ossification in AS[7,8]. Moreover, AS-MSC undergoing osteogenic differentiation exhibit dysfunction in the ability to regulate macrophages, leading to local inflammation characterized by high concentrations of TNF-α[9]. These results suggest that MSC dysfunctions are likely to be the connection between chronic inflammation and ectopic ossification in AS.

Widely known as the central inflammatory cytokine of AS, TNF-α has been proven to contribute greatly to chronic inflammation and ectopic ossification[30]. Nevertheless, whether the high concentration of TNF-α that results from AS-MSC dysfunction during osteogenesis, in turn, affects AS-MSC is still unknown. A previous study showed that TNF-α promoted MSC migration[31]. In addition, the abnormal migration ability of MSC mediated by TNF-α is involved in the pathogenesis of bone-related diseases[32,33]. In this study, we isolated AS-MSC from 15 AS patients during the active period and HC-MSC from 15 healthy donors. We found that AS-MSC had a stronger capacity for migration than did HC-MSC when cocultured with M1 macrophages and that this difference could be rectified with an anti-TNF-α neutralizing antibody but not anti-IL17 or anti-IL23 neutralizing antibodies. Moreover, in the context of stimulation with 100 ng/ml TNF-α, AS-MSC showed a stronger capacity for directional migration than HC-MSC both in vitro and in vivo. These results indicated that M1 macrophage-derived TNF-α, in turn, led to enhanced directional migration of AS-MSC.

The levels of several inflammatory cytokines including TNF-α are elevated locally in the pathological tissue of AS[34]. Notably, the

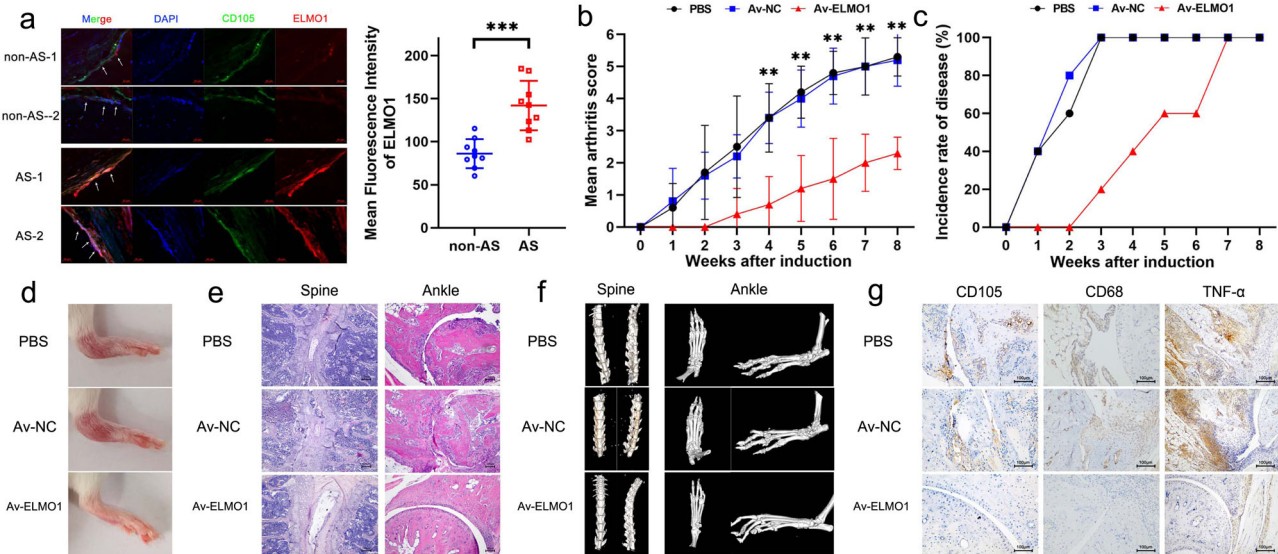

**Fig. 5 ELMO1 acts as a therapeutic target in AS. a** ELMO1 expression, quantified by mean fluorescence intensity, was higher in CD105$^+$ MSC at the enthesis site in AS patients ($n = 9$) than those at the enthesis site in non-AS patients ($n = 9$). $P = 0.000123$ based on two-tailed Student's $t$-test. Scar bar = 20 μm. **b** The mean arthritis score of the AV-ELMO1 group ($n = 5$) was lower than that of the PBS ($n = 5$) and AV-NC ($n = 5$) groups from 4 to 8 weeks. $P$ value shown in source data was based on one-way ANOVA followed by Bonferroni's post hoc comparisons. **c** The incidence rate of the AV-ELMO1 ($n = 5$) group was lower than that of the PBS ($n = 5$) and AV-NC ($n = 5$) groups before week 7. **d** The degree of ankle swelling in the AV-ELMO1 group was much slighter compared to that in the PBS and AV-NC groups. **e** H&E staining showed that tissue lesions in the spine and ankles were much slighter in the AV-ELMO1 group than in the PBS and AV-NC groups. Experiment was repeated in five mice of each group independently. Scar bar = 100 μm. **f** Micro-CT scanning showed that osteophyte formation in the spine and ankles was much less in the AV-ELMO1 group than in the PBS and AV-NC groups. **g** The numbers of migratory CD105$^+$ MSC, the numbers of CD68$^+$ macrophages and expression of TNF-α in local tissues were less in the AV-ELMO1 group than in the PBS and AV-NC groups. Experiment was repeated in five mice of each group independently. Scar bar = 100 μm. PBS group indicates SKG treated with PBS. Av-NC group indicates SKG treated with control adenovirus. Av-ELMO1 group indicates SKG treated with adenovirus encoding an shRNA specific for ELMO1. Values are presented as the mean ± SD. ** Indicates $P < 0.01$, and *** indicates $P < 0.001$.

enhanced directional migration of AS-MSC was observed only at a relatively high concentration of TNF-α (100 ng/ml), indicating a pathological rather than physiological level of TNF-α in vivo. Therefore, we suggest that this enhanced directional migration potential may exist only in local inflammatory positions with higher concentration of TNF-α, rather than in the whole body of AS patients. This speculation could be one of the reasons for specific site involvement such as enthesitis in AS.

How TNF-α mediates the enhanced directional migration of AS-MSC was the critical question we aimed to answer. To address this issue, RNA sequencing of HC-MSC and AS-MSC was performed. After treatment with 100 ng/ml TNF-α, the gene expression profiles of AS-MSC were significantly different from those of HC-MSC, with a total of 489 differentially expressed mRNAs identified. Further bioinformatic analysis determined that these 489 key mRNAs were enriched in the biological process chemotaxis and the molecular function chemokine activity, and the chemokine signaling pathway was one of the most remarkably enriched pathways identified by KEGG analysis. Therefore, although TNF-α affects various aspects of MSC function, enhanced directional migration of AS-MSC was the most significant dysfunction caused by the high concentration of TNF-α. In addition, among the 489 differentially expressed mRNAs, we determined that ELMO1 was the most important molecule with the largest fold change, which was further confirmed at the gene and protein levels, as well as at the cellular and AS patient tissue levels. ELMO1, which functions through binding to DOCKs, is a core intersectional molecule in chemotaxis and chemokine signaling pathways[13,35,36]. Specifically, ELMO1 was reported to bind to DOCK2 and then activate Rac1, which regulate intracellular actin dynamics[37]. In addition to taking part in physiological processes, ELMO1 is involved in pathological conditions such as

tumor invasion and local inflammation when abnormally expressed[15,17,18]. In our study, inhibiting ELMO1 in AS-MSC rectified the enhanced directional migration of these cells in vitro and in vivo, and elevating ELMO1 expression in HC-MSC resulted in migration dysfunction similar to that of AS-MSC. Hence, we concluded that ELMO1 is involved in the central mechanism of enhanced directional migration of AS-MSC induced by TNF-α.

Notably, several other chemokines, such as CCL4 and CCL13, were also differentially expressed in AS-MSC after TNF-α stimulation. These chemokines are reported to regulate MSC migration[38]. However, we focused on ELMO1 because it had not only the largest fold change but also a central intersectional role in the chemokine signaling pathway. This means that targeting one of these chemokines barely affected the others' functions, but regulating ELOM1 could kept within limits of most chemokines. This hypothesis was confirmed by the in vivo experiments presented in our study. Using SKG mice, the AS feature was induced by an intraperitoneal injection of curdlan, and we found that the incidence time was delayed and that the disease activity was significantly decreased after antagonizing ELMO1. In addition, local inflammation and histological changes were ameliorated in both the spine and the ankles of SKG mice after Av-ELMO1 injection. Although we could not conclude whether Av-ELMO1 affected cells other than MSC, Av-ELMO1 markedly reduced the numbers of migratory CD105$^+$ MSC and CD68$^+$ macrophages and their secretory TNF-α levels in local tissues of SKG mice. We suggested that targeting ELMO1 could restrain the enhanced migration of AS-MSC, which therefore ameliorated macrophage-mediated inflammation and subsequent pathological changes in tissues. These results not only emphasize the critical roles of ELMO1 in the enhanced directional migration of AS-MSC and

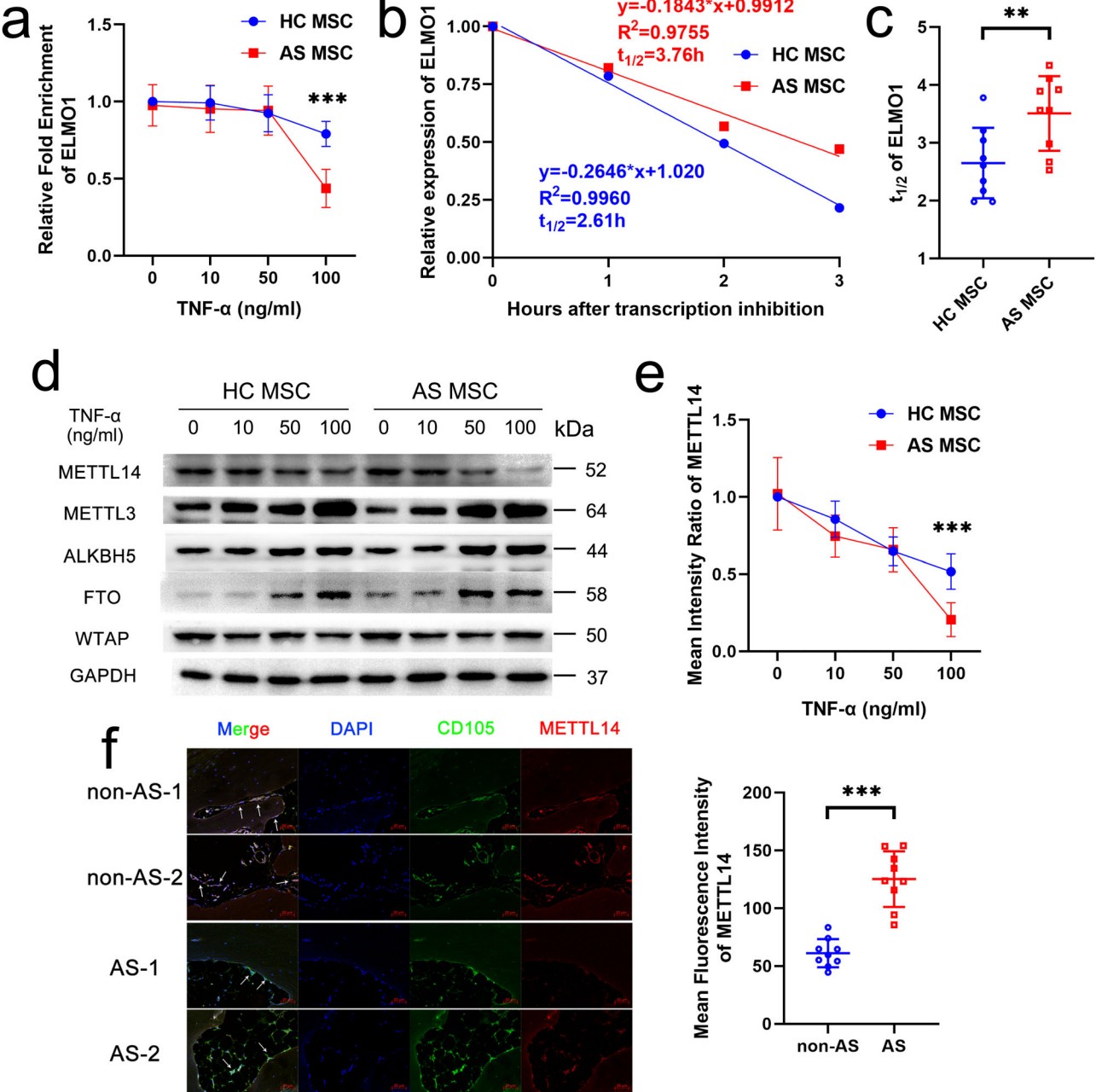

**Fig. 6 Decreasing METTL14 expression leads to lower m6A modification and RNA degradation of *ELMO1* in TNF-α-treated AS-MSC. a** The m$^6$A modification level of ELMO1 decreased after 100 ng/ml TNF-α treatment, and the m$^6$A level of *ELMO1* in AS-MSC ($n = 9$) was lower than that in HC-MSC ($n = 9$; $P = 0.000003$) with 100 ng/ml TNF-α treatment. **b** The degeneration curves of *ELMO1* mRNA for HC-MSC and AS-MSC treated with 100 ng/ml TNF-α. **c** The $t_{1/2}$ of *ELMO1* mRNA in AS-MSC ($n = 9$) was higher than that in HC-MSC ($n = 9$; $P = 0.01$) after 100 ng/ml TNF-α treatment. **d**, **e** The METTL14 expression in AS-MSC ($n = 9$) treated with 100 ng/ml TNF-α was lower than that in HC-MSC ($n = 9$; $P = 0.000024$). **f** The mean fluorescence intensity of MET was higher in CD105$^+$ MSC at the enthesis site in AS patients ($n = 9$) than those at the enthesis site in non-AS patients ($n = 9$; $P = 0.000002$). Scar bar = 20 μm. Data were analyzed using two-tailed Student's *t*-test. Values are presented as the mean ± SD. ** Indicates $P < 0.01$, and *** indicates $P < 0.001$.

the pathogenesis of AS, but also foreshadow the promising application of ELMO1 in the clinical treatment of AS.

Another key question remains: how does TNF-α induce elevated expression of ELMO1 in AS-MSC. Recent studies have shown that cytokines at relatively high concentrations can lead to abnormal changes in RNA modification and the subsequent gene expression profile, which results in tissue inflammation and autoimmune diseases[39,40]. In our study, we noticed that the increasing ELMO1 expression in AS-MSC occurred first at the mRNA level after high-

concentration TNF-α stimulation, suggesting that RNA modification may be involved in this issue. Recently, TNF-α was determined to inhibit MSC differentiation by mediating m$^6$A modification of specific mRNAs[41]. In addition, m$^6$A modification regulates gene expression and thus affects cell migration[42]. m$^6$A, the most abundant internal modification on mRNA, can accelerate mRNA degeneration and negatively affect gene expression and is widely involved in cell function and development[19–21]. Furthermore, abnormal m$^6$A modification plays important roles in the

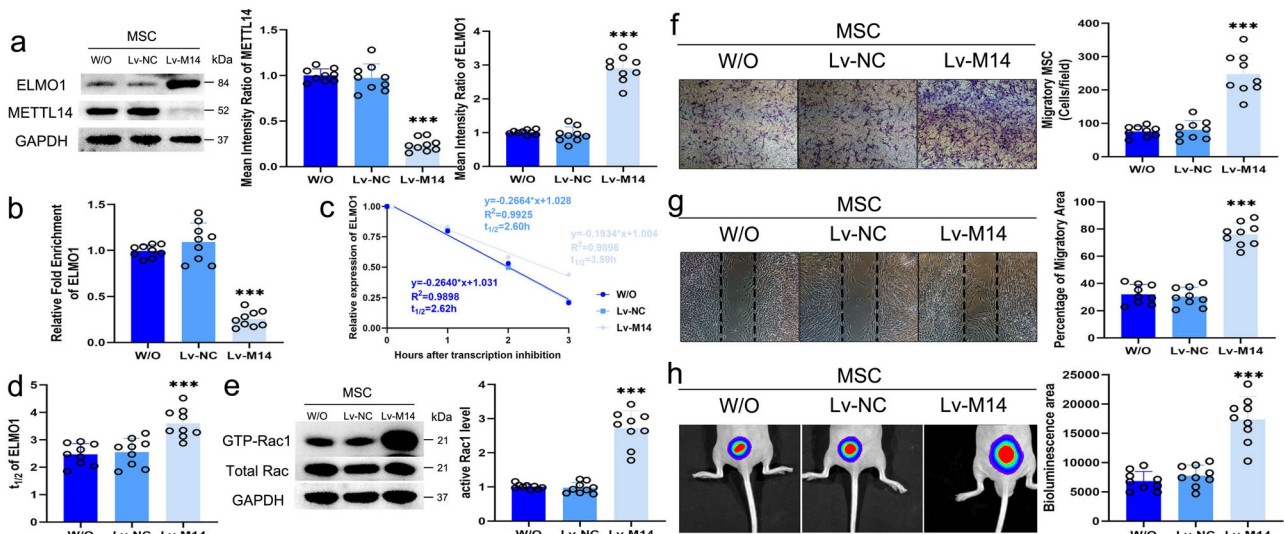

**Fig. 7 Inhibiting METTL14 expression increases ELMO1 expression and promotes MSC directional migration. a** METTL14 expression in MSC was lower in Lv-M14 group ($n = 9$; $P = 7.2995E-14$), and ELMO1 expression was higher in Lv-M14 group compared to Lv-NC group ($n = 9$; $P = 1.3076E-13$). **b** The m6A modification level of Lv-M14-transfected MSC ($n = 9$) was lower compared to that of control MSC ($n = 9$; $P = 3.0324E-11$) and Lv-NC-transfected MSC ($n = 9$; $P = 2.7411E-12$). **c**, **d** The degeneration rate of *ELMO1* mRNA in Lv-M14 group ($n = 9$) was slower than that in control ($n = 9$; $P = 0.00013$) and Lv-NC group ($n = 9$; $P = 0.000331$). **e** The Lv-M14 group ($n = 9$) had higher active Rac1 levels than control group ($n = 9$; $P = 8.9659E-11$) and Lv-NC group ($n = 9$; $P = 6.3663E-11$). **f** There were more stained migratory MSC in the Lv-M14 group ($n = 9$) than in the control group ($n = 9$; $P = 7.6189E-9$) and Lv-NC group ($n = 9$; $P = 1.3407E-8$). **g** The migratory area of the Lv-M14 group ($n = 9$) was larger than that of the control group ($n = 9$; $P = 1.7401E-11$) and Lv-NC group ($n = 9$; $P = 7.5404E-12$). **h** The bioluminescence area of the Lv-M14 group ($n = 9$) was larger than that of the control group ($n = 9$; $P = 4.093E-8$) and Lv-NC group ($n = 9$; $P = 2.0632E-7$). Data were analyzed using one-way ANOVA followed by Bonferroni's post hoc comparisons. Values are presented as the mean ± SD. *** Indicates $P < 0.001$. W/O indicates MSC without lentivirus transfection. Lv-NC indicates MSC transfected with control lentiviruses. Lv-M14 indicates MSC transfected with lentiviruses encoding an shRNA specific for METTL14.

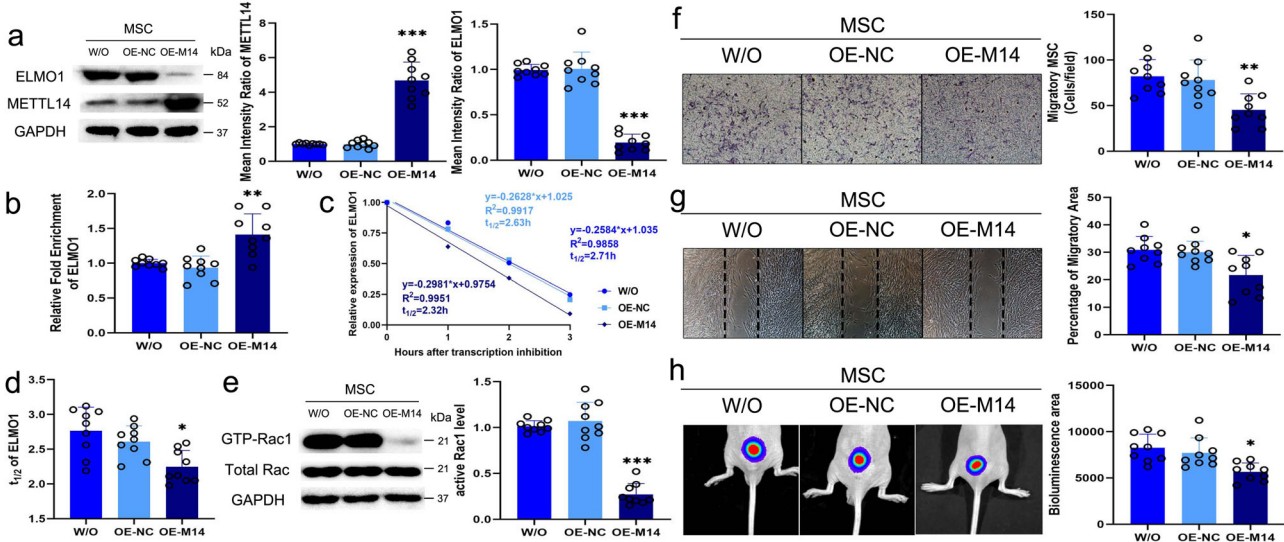

**Fig. 8 METTL14 overexpression downregulates ELMO1 levels and inhibits MSC directional migration. a** METTL14 expression in MSC was increased and ELMO1 expression was inhibited in the OE-M14 group ($n = 9$) compared to the control group ($n = 9$; $P = 1.3363E-11$, $8.9609E-13$) and OE-NC group ($n = 9$; $P = 1.2451E-11$, $7.4978E-13$). **b** The m6A modification level of *ELMO1* mRNA in OE-M14 group ($n = 9$) was higher than that of control group ($n = 9$; $P = 0.001$) and OE-NC group ($n = 9$; $P = 0.000092$). **c**, **d** The degeneration rate of *ELMO1* mRNA was elevated by OE-M14 group ($n = 9$) than OE-NC group ($n = 9$; $P = 0.029$). **e** The active Rac1 level in MSC ($n = 9$) was significantly lower in OE-M14 group ($n = 9$; $P = 1.3512E-10$) than the OE-NC group ($n = 9$; $P = 2.0426E-11$). **f** The number of stained migratory MSC in OE-M14 group ($n = 9$) was lower than that of control group ($n = 9$; $P = 0.001$) and OE-NC group ($n = 9$; $P = 0.005$). **g** The migratory area of the OE-M14 group ($n = 9$) was smaller compared to that of control group ($n = 9$; $P = 0.005$) and OE-NC group ($n = 9$; $P = 0.011$). **h** The bioluminescence area of nude mice in the OE-M14 group ($n = 9$) was lower than that of mice in the control group ($n = 9$; $P = 0.002$) and the OE-NC group ($n = 9$; $P = 0.013$). Data were analyzed using one-way ANOVA followed by Bonferroni's post hoc comparisons. Values are presented as the mean ± SD. * Indicates $P < 0.05$, ** indicates $P < 0.01$, and *** indicates $P < 0.001$. W/O indicates MSC without lentivirus transfection. OE-NC indicates MSC transfected with control lentiviruses. OE-M14 indicates MSC transfected with lentiviruses overexpressing METTL14.

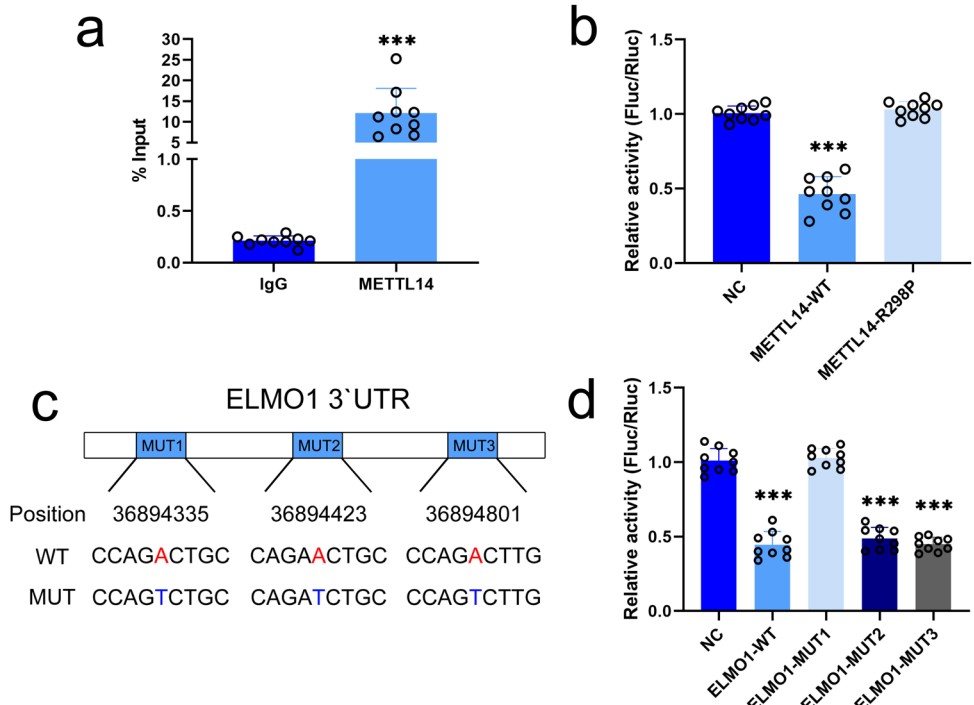

**Fig. 9 METTL14 acts on the specific m6A modification site of the *ELMO1* 3′UTR. a** CLIP-PCR assay results showed that *ELMO1* expression was much higher in the anti-METTL14 antibody group ($n = 9$ biologically independent cells) than in the IgG control group ($n = 9$ biologically independent cells). $P = 0.000018$ based on two-tailed Student's *t*-test. **b** Luciferase activity was lower in the wild-type METTL14 group ($n = 9$ biologically independent experiments) compared to the R298P mutant METTL14 group ($n = 9$ biologically independent experiments; $P = 6.5606E{-}14$) and NC group ($n = 9$ biologically independent experiments; $P = 2.26E{-}13$). Data were analyzed using one-way ANOVA followed by Bonferroni's post hoc comparisons. **c** The mutated sites (A to T) in the *ELMO1* 3′UTR. **d** The luciferase activity of the mutant 1 group ($n = 9$ biologically independent experiments) was restored to control levels, but the luciferase activity of the mutant 2 ($n = 9$ biologically independent experiments; $P = 1.7615E{-}18$) and 3 groups ($n = 9$ biologically independent experiments; $P = 2.6302E{-}17$) as well as the wild-type ELMO1 group ($n = 9$ biologically independent experiments; $P = 2.1376E{-}18$) were significantly lower. Values are presented as the mean ± SD. *** Indicates $P < 0.001$ compared to the IgG or NC group.

pathogenesis of rheumatic diseases, including rheumatoid arthritis and systemic lupus erythematosus[43,44]. After treatment with 100 ng/ml TNF-α, the m6A level of *ELMO1* in AS-MSC was obviously lower than that in HC-MSC. In addition, the degeneration rate of *ELMO1* mRNA in AS-MSC was slower than that in HC-MSC, which was consistent with the increasing level of *ELMO1* mRNA expression in AS-MSC after TNF-α stimulation. Herein, we demonstrated the critical role of m6A modification in the elevated expression of ELMO1 and the subsequent enhanced directional migration of AS-MSC.

m6A modification is regulated by specific methyltransferases and demethylases[45]. These m6A-related proteins participate in the regulation of MSC functions[22,41]. Herein, although the expression of most m6A-related proteins was altered after TNF-α stimulation, a difference between AS-MSC and HC-MSC was found for only METTL14 after TNF-α treatment at a concentration of 100 ng/ml. Additionally, METTL14 expression in MSC from AS enthesis tissue was abnormally decreased compared with that from the control tissue. Previously, METTL14, an m6A methyltransferase, was proven to regulate breast cancer cell migration through m6A modification[46]. Furthermore, inhibiting METTL14 in MSC decreased the m6A level of *ELMO1* and then upregulated ELMO1 expression at the protein level, thereby promoting the migratory ability of MSC in vitro and in vivo. Overexpressing METTL14 produced inverse results in MSC, confirming that METTL14 negatively regulates ELMO1 expression through m6A modification in MSC stimulated with TNF-α. Therefore, METTL14 was the key mediator of ELMO1 leading to enhanced directional migration of AS-MSC.

m6A modification regulates specific gene expression by increasing or decreasing mRNA stability. The different effect of m6A modification depends on the predominant m6A methyltransferases or demethylases and their binding m6A reader enzymes[47]. A previous study demonstrated that METTL14 affected the mRNA stability of *MYB* and *MYC* through m6A modification, which ultimately inhibited hematopoietic stem cell differentiation[26]. In our study, we found that the *ELMO1* mRNA half-life was negatively correlated to its m6A level and the METTL14 expression level. Additionally, in our experiment system, we determined that METTL14 mainly bound YTHDF2 and YTHDF3, which are the two m6A reader enzymes mainly promoting mRNA degradation[47]. We concluded that METTL14 along with YTHDF2 and YTHDF3 decreased the *ELMO1* mRNA stability and increased its degradation rate after m6A modification, which led to a lower expression of ELMO1 in the TNF-α treated MSC.

As features distinguishing AS from other rheumatic diseases, chronic inflammation and ectopic ossification in entheses are the central pathological features of AS that exist outside the bone[2]. However, it is well known that MSC are located in the bone marrow. Although we have demonstrated that multiple dysfunctions in AS-MSC contribute to the pathogenesis of AS[7,9], it is still unclear how MSC migrate to and participate in enthesis lesions. As normal MSC migrate to certain sites and participate in tissue regeneration and immune regulation (42), the migration of MSC with abnormal capacities result in pathological lesions and contributes to the development of diseases, such as rheumatoid arthritis[48]. Therefore, it is important to illuminate the migratory

ability of AS-MSC under certain conditions. Previously, microscopic holes connecting entheses with the bone marrow were observed in the cortical bone, and bone marrow tissues were shown to spill into entheses through these holes, which provides the anatomical basis for MSC migration[2,49]. In this study, we demonstrated that under high-concentration TNF-α stimulation, AS-MSC exhibited an enhanced directional migration ability facilitated through METTL14-mediated m6A modification of *ELMO1*. Summarizing our previous and current results[7,9], we suggest that a vicious three-step cycle that is hard to break exists in the entheses of AS patients: (1) TNF-α at a relatively high concentration in the enthesis induces the directional migration of AS-MSC, recruiting AS-MSC from the bone marrow to the site of enthesis. (2) These migratory AS-MSC show an enhanced osteogenesis capacity to be involved in ectopic ossification. (3) AS-MSC subsequently secrete a large amount of CCL2 during the osteogenic differentiation course, which augments monocyte migration, increases proinflammatory macrophage polarization and enhances TNF-α secretion at the site of enthesis. Despite controversy regarding the relationship between chronic inflammation and ectopic ossification in AS[50], our study adds support to the close relationship of these two features through the vicious cycle of MSC and TNF-α. However, current available evidence does not reveal the origin of this vicious cycle, but a study showing that early application of TNF-α blockers inhibits spinal radiographic progression in AS implies a possible upstream role for TNF-α[51]. Recent studies have indicated that various factors, including mechanical strain and the gut microbiota, may trigger the elevation in TNF-α levels[2,52], but this needs further investigation.

Our study may provide insight for the clinical treatment of AS. Successfully breaking the above-mentioned vicious cycle by targeting key molecules could markedly improve AS. Based on our in vivo experiments, we found that targeting ELMO1 could significantly ameliorate the clinical manifestations and tissue lesions in SKG mice. Recently, a small molecule inhibitor of DOCK named CPYPP was used to antagonize ELMO1/DOCK Rac1 GEF activity[53]. This inhibitor targeting ELMO1/DOCK/Rac1 showed great ability in releasing inflammation[54,55], and may have great application potential in AS. The therapeutic effect of ELMO1 requires further studies in the future.

In summary, we demonstrated that TNF-α accelerated the directional migration of AS-MSC through METTL14-dependent m6A modification of *ELMO1*. This study may contribute to understanding not only the pathogenesis but also the diagnosis and treatment of AS. Some issues, such as the trigger of the vicious cycle and the sequential order of inflammation and ossification, remain unclear and need to be addressed in the future.

## Methods

**Study approval**. This study conforms to the Declaration of Helsinki and was approved by the Ethics Committee of the Eighth Affiliated Hospital, Sun Yat-Sen University, Guangzhou, China. The experiments on mice were approved by the Institutional Animal Care and Use Committee of Sun Yat-Sen University, Guangzhou, China. All experimental procedures on mice were carried out in strict adherence to the rules and guidelines for the ethical use of animals in research.

**Cell isolation and culture**. A total of 15 patients with AS and 15 healthy controls were recruited for this study. All the AS patients were diagnosed according to the 1984 New York modified criteria[56]. After being informed of the possible risks and complications of bone marrow puncture, all the AS patients and healthy controls signed informed consent forms. The characteristics of the study subjects are presented in Supplementary Table 1. MSC were immediately isolated and purified from bone marrow samples using density gradient centrifugation. MSC were resuspended in Dulbecco's modified Eagle's medium supplemented with 10% fetal bovine serum and then seeded into 25-cm$^2$ flasks and cultured at 37 °C in 5% CO$_2$. The cells in suspension were removed, and the medium was replaced every 3 days

thereafter. When the culture reached 90% confluence, MSC were digested using 0.25% trypsin containing 0.53 mM EDTA and reseeded in new flasks; these cells were expanded and used for experiments.

HEK293T cells were cultured in high-glucose DMEM supplements with 10% FBS at 37 °C under 5% CO$_2$. At 80–90% confluence, HEK293T cells were digested with 0.25% trypsin containing 0.53 mM EDTA and reseeded in new flasks.

**MSC migration assays in a Transwell system**. MSC migration assays were performed using Polycarbonate Membrane Transwell® Inserts (8.0-μm pores, 24-well plate; Corning). Briefly, a total of 2 × 10$^4$ MSC in 100 μl DMEM were seeded in the upper chambers, and 600 μl DMEM without or with TNF-α at a concentration ranging from 10 to 100 ng/ml was added to the lower chambers. In the assays of Fig. 1a and Supplementary Figs. 1, 2 × 10$^5$ macrophages differentiated from CD14$^+$ monocytes isolated from peripheral blood samples were suspended in 600 μl DMEM and then seeded in the lower chambers without or with 0.5 μg/ml anti-TNF-α, 0.05 μg/ml anti-IL-17 or 0.5 μg/ml anti-IL-23 neutralizing antibody. After coculturing for 24 h, the upper chambers were washed three times with phosphate-buffered saline (PBS) and fixed with 4% paraformaldehyde. Then, the upper chambers were stained with 0.1% crystal violet for 15 min. The migratory MSC on the lower side of the chamber were photographed and counted as the mean number of cells per 10 random fields for each chamber in two separate experimenters.

**Wound-healing assays**. Wound-healing assays were performed using IBIDI® culture-Inserts 2 Well for self-insertion. Each insert was placed in a well of a 12-well plate. MSC (5 × 10$^4$) in 200 μl DMEM were seeded in a well of the insert. The inserts were then removed, and DMEM without or with TNF-α at a concentration ranging from 10 to 100 ng/ml was added. After 12 h of migration, the migratory area was calculated using ImagePro Plus 6.0. The percentage of the migratory area was defined as the ratio of the MSC migratory area at 12 h to the primary wound area created by the insert at 0 h.

**μ-slide chemotaxis assay**. Directional migration assays were performed using IBIDI® μ-slide chemotaxis chambers. An MSC suspension (6 μl, 3 × 10$^6$ cells) was seeded into the center channel of μ-slide chambers. After 12 h of incubation, DMEM containing TNF-α was added to the left medium reservoir, and DMEM without TNF-α was added to the right one reservoir. The cell migration track was imaged every 10 min over 24 h with the BioTek Lionheart™ FX Automated Live Cell Imager with Augmented Microscopy™. Cell tracking analysis was performed using the Manual Tracking plugin for ImageJ 1.51 and the Chemotaxis and Migration Tool according to the IBIDI protocol.

**Cell immunofluorescence assay**. For the cell immunofluorescence assay, 5 × 10$^4$ MSC in 1 ml DMEM were seeded in each well of a 12-well plate and then treated with or without 100 ng/ml TNF-α for 24 h. Then, the cells were washed with PBS, fixed with 4% paraformaldehyde and incubated with 1% Triton X-100. MSC were blocked in goat serum and then incubated with an anti-ELMO1 antibody (1:500) overnight at 4 °C. The cells were washed and incubated with a fluorescein-labeled secondary antibody (Alexa Fluor® 647; 1:3000) for 1 h, followed by staining with DAPI (Thermo Fisher) for 10 min. All images were obtained using an LSM 5 Exciter confocal imaging system (Carl Zeiss).

**In vivo migration assay**. MSC were transfected with a lentivirus encoding luciferase (OBiO Technology) as described below (MSC/Fluc). Mice were housed at the Laboratory Animal Center of Sun Yat-Sen University under specific pathogen-free conditions, with a 12-h light/dark cycle in a temperature ((22 ± 2 °C)) and humidity-controlled room (60%) with free access to water and food. Eight-week-old BALB/c-nu/nu female mice (Laboratory Animal Center of Sun Yat-Sen University) were anesthetized via intraperitoneal injection of ketamine and xylazine. A total of 5 × 10$^5$ MSC/Fluc in 20 μl DMEM were injected subcutaneously into the dorsal sides of the BALB/c-nu/nu mice. Twenty microliters of TNF-α at a concentration of 500 ng/ml was administered at four sites around the MSC injection point at a distance of 0.5 cm. The injected MSC/Fluc were observed using a Xenogen IVIS Spectrum system (Caliper Life Sciences, Inc.) on days 0, 1, 3, and 5 after intraperitoneal injection of 3 mg D-Luciferin potassium. The bioluminescence area was analyzed using Living Image 2.12 and ImagePro Plus 6.0. On day 7 after injection, the mice were sacrificed, and the tissues at the TNF-α injection sites were obtained for immunohistochemical analysis as described below.

**RNA extraction, reverse transcription, and quantitative real-time PCR**. Total RNA was isolated from MSC using TRIzol and transcribed into cDNA using a PrimeScript™ RT reagent kit according to the protocols. Quantitative real-time PCR was performed on the LightCycler®480 PCR System (Roche) using TB Green® Premix Ex Taq™. The relative expression levels of each gene were analyzed using the 2$^{-\triangle\triangle Ct}$ method. The forward and reverse primers for each gene are shown in Supplementary Table 2.

**RNA sequencing and data analysis**. MSC were plated and treated with or without 100 ng/ml TNF-α for 24 h. RNA was extracted as described above. cDNA library construction and sequencing were performed by the Beijing Genomics Institute using the BGISEQ-500 platform. The sequencing data were filtered with SOAPnuke 1.5.2, and the clean reads were mapped to a reference genome using HISAT2 2.0.4. After alignment using Bowtie2 2.2.5, the expression level of each gene was calculated by RSEM 1.2.12, and differential expression analysis was performed using DESeq2 1.4.5 with the parameters fold change ≥2 and adjusted $P$ value ≤0.001. The sequencing data analysis, including heatmap clustering, principal component analysis (PCA), Venn diagram creation, gene ontology (GO) analysis, and Kyoto Encyclopedia of Genes and Genomes (KEGG) analysis, were performed using BGI Dr. Tom 2.0.

**Protein extraction and western blot**. MSC were lysed in RIPA lysis buffer containing protease inhibitors and phosphatase inhibitors. The lysates were centrifuged at $1100 \times g$ at 4 °C for 30 min, and the lysate protein concentrations were determined using a Pierce BCA protein assay kit. The immunoprecipitants was obtained in Co-IP assay as described below. After boiling with sample loading buffer, equal amounts of protein extracts or the immunoprecipitants were separated using sodium dodecyl sulfate-polyacrylamide gel electrophoresis and subsequently transferred to polyvinylidene fluoride (PVDF) membranes (Millipore). The PVDF membranes were blocked and incubated overnight at 4 °C with primary antibodies against GAPDH, ELMO1, ELMO2, ELMO3, METTL3, METTL14, FTO, ALKBH5, WTAP, DOCK1, DOCK2, DOCK4, DOCK5, DOCK8, or Rac1 (1:1000). After washing three times, the PVDF membranes were incubated with a horseradish peroxidase (HRP)-conjugated secondary antibody (1:3000). Specific antibody-antigen complexes were detected using Immobilon Western Chemiluminescent HRP Substrate. The mean intensity ratio was determined and analyzed using a UVP ChemStudio PLUS system and ImagePro Plus 6.0.

**Rac1 activation assay**. Rac1 activation in MSC was detected with the Active Rac1 Detection Kit through a GST pulldown assay according to the kit protocol. Briefly, protein was extracted from MSC treated with or without TNF-α as described above. Glutathione resin containing agarose beads was added to the spin cup with a collection tube and washed three times. GST-Human PAK1-PBD was added to the spin cup, followed by protein lysis and incubation for 1 h at 4 °C with gentle rocking. After washing, the spin cup containing agarose beads was incubated with reducing sample buffer at room temperature for 2 min, and the eluted sample was collected by centrifugation. GTP-Rac1 in the eluted protein fraction was detected using Western blot analysis as described above. MSC-derived protein without GST pulldown was used to detect the total Rac1 and GAPDH levels.

**Coimmunoprecipitation (Co-IP) and liquid chromatograph (LC)-mass spectrometry (MS)/MS**. The Co-IP assay was performed using the Dynabeads™ Protein A Immunoprecipitation Kit according to the kit protocol. Briefly, an MSC protein extract was incubated with an anti-ELMO1 antibody (1:100) or IgG control at 4 °C overnight, followed by the addition of protein G agarose beads and incubation for another 4 h. Then, the immunoprecipitants were collected for LC-MS/MS and western blotting. LC-MS/MS was performed by Shanghai Applied Protein Technology Co., Ltd. Data were analyzed using Proteome Discoverer 1.4 against the UniProt database (https://www.uniprot.org/).

**Lentivirus construction and transfection**. Three ELMO1, METTL14, DOCK1, DOCK4, DOCK5, YTHDF2, and YTHDF3-specific siRNAs were designed and synthesized by OBiO Technology. The sequences are shown in Supplementary Table 3. The siRNA was used to transfect MSC using a Lipofectamine RNAi MAX according to the protocol, and the siRNA with best knockdown efficiency was chosen to construct a lentivirus encoding a short hairpin RNA (shRNA) specific for ELMO1 (Lv-ELMO1) or METTL14 (Lv-M14) by OBiO Technology. ELMO1- and METTL14-overexpression lentiviruses (OE-ELMO1 and OE-M14, respectively) and their vector controls were also constructed by OBiO Technology. Lentiviruses ($10^9$ TU/mL) and 5 μg/mL polybrene were added to DMEM and incubated with MSC for 24 h at a multiplicity of infection of 50. Experiments were performed on day 4 after transfection. Adenoviruses encoding an shRNA specific for ELMO1 (Av-ELMO1) or control adenoviruses (Av-NC) for mouse experiments were also constructed and purchased from OBiO Technology.

**Cell proliferation assay**. MSC transfected with Lv-ELMO1 (or OE-ELMO1) and its NC control were seeded in 96-well plates. Cell proliferation ability was detected using Cell Counting Kit-8 according to the protocol. Medium without cells were used as negative controls.

**$m^6A$ RNA immunoprecipitation (RIP)**. The $m^6A$ RIP assay was performed using the Magna MeRIP™ $m^6A$ Kit according to the manufacturer's instructions. Briefly, RNA was extracted and then chemically fragmented into fragments of 200 nucleotides or less. The RNA fragments were incubated with anti-$m^6A$ antibody- or IgG-conjugated Protein A/G magnetic beads at 4 °C overnight. Then, the magnetic beads were collected, and the bound $m^6A$-modified RNA was eluted for qRT-PCR analysis as described above. Equal amounts of nonimmunoprecipitated

RNA fragments were used as the input control. The fold enrichment of each target gene was calculated with the formulas below.

$$\Delta CT_{\text{target gene}} = CT_{\text{target gene}} - CT_{\text{Input}}$$
$$\Delta CT_{\text{IgG}} = CT_{\text{IgG}} - CT_{\text{Input}}$$
$$\Delta\Delta CT = \Delta CT_{\text{target gene}} - \Delta CT_{\text{IgG}}$$
$$\text{Fold enrichment} = 2^{-\Delta\Delta CT}$$

The relative fold enrichment was calculated by normalizing to the fold enrichment of HC-MSC without TNF-α stimulation.

**RNA stability assays**. MSC were seeded in a 12-well plate and treated with actinomycin D at a concentration of 20 μg/ml for 0, 1, 2, and 3 h. After treatment, MSC RNA was immediately extracted, and qRT-PCR assays were performed as described above. The turnover rate and half-life of each target gene were calculated as described in a previous study[57].

**Cross-linking and RNA immunoprecipitation (CLIP) assay**. MSC were treated with 150 mJ/cm$^2$ UVC irradiation at 254 nm for 40 s. The CLIP assay was then performed using the EZ-Magna RIP™ RNA-Binding Protein Immunoprecipitation Kit according to the manufacturer's instructions. Briefly, MSC were lysed using RIP lysis buffer, and the lysate was incubated with antibodies against METTL14, YTHDC1, YTHDC2, YTHDF1, YTHDF2, YTHDF3 (1:50) or IgG control with protein A/G magnetic beads at 4 °C overnight. The magnetic beads were immobilized, and then the immunoprecipitant containing specific protein with its bound RNA was eluted and treated with proteinase K. The remaining RNA was extracted for qRT-PCR analysis as described above. Equal amounts of non-immunoprecipitated RNA fragments were used as the input control. The %Input was calculated with the formulas below.

$$\Delta CT_{\text{target gene or IgG}} = CT_{\text{target gene or IgG}} - CT_{\text{Input}}$$
$$\%\text{Input} = 2^{-\Delta CT}$$

**Dual-luciferase reporter assay**. A METTL14 expression vector and its mutant vector (METTL14-R298P), as well as luciferase reporter vectors containing the C-terminal DNA fragment of ELMO1 or its mutant with mutations in $m^6A$ modification sites (A replaced by T), were synthesized by OBiO Technology. The possible $m^6A$ modification sites were predicted by the m6AVar database[58]. HEK293T cells were seeded in 12-well plates, and then 1.5 μg luciferase reporter vector, 1.5 μg METTL14 or its mutant vector, and 1.5 μg Renilla luciferase reporter vector were cotransfected into HEK293T cells using the Lipofectamine 3000 Transfection Kit according to the manufacturer's instructions. After 48 h of transfection, the luciferase activity was detected using the Dual-Luciferase® Reporter Assay System Kit (Promega) according to the protocol. Relative Fluc/Rluc activity was calculated by normalizing the activity of firefly luciferase to that of Renilla luciferase.

**Mouse induction, treatment and, scoring**. Male SKG mice were purchased from CLEA Japan, Inc. and housed as described above. All mice were handled in accordance with the guidelines for animal care approved by the Institutional Animal Care and Use Committee of Sun Yat-Sen University. Disease was induced at 8 weeks of age using 3 mg curdlan administered by intraperitoneal injection. The SKG mice were randomly divided into three groups: a PBS group, an Av-NC group, and an Av-ELMO1 group. The SKG mice in the Av-ELMO1 group were treated with $5 \times 10^{10}$ Av-ELMO1 via intravenous tail vein injection at the time of disease induction, and the SKG mice in the Av-NC group or PBS group were separately treated with equal amounts of control adenoviruses or PBS. Clinical features in the mice were monitored weekly by two independent observers who were blinded to the treatment groups. The score criteria were defined as previously reported[59]: 0 = no swelling or redness, 0.1 = swelling or redness of the digits, 0.5 = mild swelling and/or redness of the wrist or ankle joints, and 1 = severe swelling of the larger joints. At week 8 after induction and treatment, the mice were sacrificed, and the tissues were obtained for micro-CT examination, hematoxylin and eosin (H&E) staining, immunohistochemical analysis, and immunofluorescence as described below.

**Micro-CT scanning**. Micro-CT was performed to analyze the structures of the spine and ankle. Obtained tissues were fixed with 4% polyoxymethylene and then scanned using a Siemens Inveon CT scanner with a resolution of 19 μm. The figure data were analyzed using RadiAnt DICOM Viewer 5.0.2 software.

**H&E staining**. Harvested spine and ankle tissue samples were fixed with 4% polyoxymethylene for 24 h, decalcified in 20% EDTA for 14 days and then embedded in paraffin for sectioning. The sections were stained with hematoxylin for 5 min. After washing with PBS for 10 min, the sections were stained with eosin for 3 min. Then, the sections were dehydrated and observed using a microscope.

**Tissue immunohistochemical assay**. For the SKG mouse tissue immunohisto-chemical assay, sections were incubated in 10 mM citrate buffer and microwaved at 750 W for 30 min for antigen retrieval. The sections were treated with 3% $H_2O_2$ for 20 min and blocked with 5% normal goat serum for 1 h. Then, the sections were separately incubated with anti-CD68, anti-TNF-α or anti-CD105 (1:100) antibodies at 4 °C overnight. Secondary antibody (1:500) incubation and color development were performed using the SP Rabbit & Mouse HRP DAB Kit according to the kit protocol. For mouse tissue from the in vivo migration assay, the immunohisto-chemical assay was performed as described above except that an antibody specific for human HLA Class 1 ABC was used.

**Tissue immunofluorescence assay**. For the human enthesis tissue immuno-fluorescence assay, nine AS patients and 9 non-AS patients were recruited. Sites of ossifying enthesis were confirmed via presurgical image analyses and visual observations during surgery. Ossifying tissues were obtained during lumbar spine surgeries. The characteristics of the study subjects are presented in Supplementary Table 4. The tissue samples obtained were successively fixed, decalcified, and embedded in paraffin. Sections were deparaffinized, hydrated, and incubated in 1% Triton X-100/PBS. After antigen retrieval in citrate buffer and blocking in goat serum, the sections were incubated with anti-CD105, anti-ELMO1 or anti-METTL14 antibodies (1:100) overnight at 4 °C. The sections were washed and incubated with a fluorescein-conjugated secondary antibody (1:400) for 1 h and then with DAPI for another 10 min. All images were obtained using an LSM 5 Exciter confocal imaging system (Carl Zeiss). The mean fluorescence intensity was quantified using the ImagePro Plus 6.0.

**Statistical analysis**. All the results were determined based on at least three separate in vitro experiments containing at least triplicate samples. The Shapiro–Wilk normality test was used to check the normality of the data, and data with a Shapiro–Wilk test P > 0.05 were considered to fit a normal distribution. The two group comparisons were performed using a 2-tailed Student's t-test, and comparisons of three or more different groups were performed by a one-way ANOVA, followed by Bonferroni's post hoc comparisons. Data are expressed as the means±standard deviations. Statistical analysis was performed with SPSS (SPSS Inc.). The n values indicate the numbers of individuals in each experiment. P-values less than 0.05 were considered statistically significant.

**Reporting summary**. Further information on research design is available in the Nature Research Reporting Summary linked to this article.

## Data availability
The data of RNA sequencing generating in this study has been deposited in the NCBI BioProject database under accession code PRJNA700800. UniProt database (https://www.uniprot.org/) was used in the study. All other data supporting the findings of this study are available within the article and its supplementary data. Source data are provided with this paper.

## Code availability
No codes have been developed for this project.

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

## Acknowledgements

The reagent information was shown in Supplementary Table 7. This study was financially supported by the National Natural Science Foundation of China (81971518), the Key-Area Research and Development Program of Guangdong Province (2019B020236001), the Guangdong Basic and Applied Basic Research Foundation (2020A1515010097), the Fundamental Research Funds for the Central Universities (19ykpy01) and the Public Health and Welfare Research Project of Futian district of Shenzhen (FTWS2019020).

## Author contributions

Z.X., P.W., Y.W. and H.S. contributed to designing research studies. Z.X., W.Y., G.Z., W.L., M.L., J.L., P.W., Y.W. and H.S. contributed to analyzing data. Z.X., W.Y., G.Z., J.L., S.C., Z.L., G.Y., Z.S., Y.C. and F.Y. contributed to conducting experiments. Z.X., W.Y. and G.Z. contributed to writing the manuscript.

## Competing interests

The authors declare no competing interests.
