## [Peer Review File · Nature Communications]

Reviewer comments, first round –

Reviewer #1 (Remarks to the Author):

In this manuscript, the authors aim to define the mechanisms whereby TNF α could be contributing to ankylosing spondylitis (AS). They define that high dose TNF α increases migration of AS-MSC in cell and graft models in vivo. By conducting transcriptomics experiments, they identify a large network of modulated genes. They focus on Elmo1 since it is integrating many pro-migratory cytokine receptor pathways. Depletion of Elmo1 decreases migration in cells as well as in vivo in graft assay (TNF α -induced). A central experiment is the in vivo depletion of Elmo by systematic siRNA injection in a model of SGK mice that appears to decrease arthritis. Next, the authors investigated the mechanisms of Elmo1 over expression in AS-MSC cells. They identified the m6A modifier MTTL14. Depletion of MTTL14 increases Elmo1 expression while its over-expression decreased the stability of the Elmo mRNA. They identified a region in the Elmo UTR that is sensitive to MTTL14 activity.

Overall, the data presented is technically solid. This is the biggest strength of the manuscript. However, it fails to bring a major mechanistic contribution given that (1)-Elmo proteins have already been tightly linked to increased migration, (2) knockout of Elmo has been reported to dampen the features of arthritis and (3) the MTTL14 story is poorly connected to the rest of the story. The cell biology presented in this paper is also very preliminary and over-interpreted.

MAJOR COMMENTS.

1. The patient data is barely discussed in the manuscript, but instead in the Methods section. This is quite unusual and not appropriate for such a manuscript where this info should be clearly presented in the main text.

2. The authors claim that Elmo1 promotes, and localizes to, filopodia in AS-MSC. This is very poorly conducted in the manuscript. First the Elmo-Dock-Rac signaling axis would be more consistent with lamellipodia formation. This is not discussed. Also, naming cellular extensions as "filopodia" is premature without proper staining for markers of these structures. This data should all be removed if not analyzed correctly as it fails to provide any new insights and is largely overinterpreted.

3. Elmo1 is already tightly connected to cell migration. In this manuscript, the authors fail to provide any real insights on how it might function other than promote Rac activation. Even the Rac connections is not formally demonstrated. One would assume Elmo1 signals via Dock proteins (1,4 and 5 expressed in their cells). This should at a minimum be validated with siRNAs. The authors provide proteomics data on Elmo1 immunoprecipitates in +/- TNF α to attempt to make new connections (I assume), but this only reveals known interactors. This data should clearly be removed from this manuscript. In a high-profile journal, if the data is not used or dissected correctly, it is not useful. One idea would have been to begin connecting a novel chemokine receptor upstream of Elmo. The authors mention that there could be redundancy among those, but it has not been tested really. This may in fact uncover yet another clinical intervention point.

4. I have a major issue with the mouse model used as a pre-clinical model. First of all, it is poorly presented in the manuscript, even misleading to the reader. The SGK mice were treated to Curdlan to induce the features of AS (as done by others in the literature). This is not mentioned at all but in the methods. Most importantly, the authors did not analyze the expression of Elmo in this model. Is Elmo induced? This is critical since if it is Elmo-independent, then the knockdown of Elmo1 by siRNA may be of pleiotropic origin rather than really targeting the vicious cycle studied here (TNF α , Elmo1 expression). The model must be validated as a model useful to assess the contribution of Elmo1. Also, why didn't the author cross Elmo1 -/- mice with their SGK mouse model? This may have been a cleaner genetic experiment. Finally, it is never discussed clearly how

Elmo1 might be targeted in vivo. One guess would be that the Dock protein would be the ideal target as inhibitors have been developed of the years (Elmo being a scaffold).

5. The MTTL14 story is potentially very interesting but appears randomly in the study. Did the authors find any evidence in their RNA-seq data that MTTL14 expression is decreased in AS-MSC treated with TNF? I fail to see any "biological process" or "machinery" in the presented data that would suggest it is the case. What about in the patient samples?

6. While the MTTL14 data is per se clean and convincing, the authors completely overlook the fact that interfering with MTTL14 may have pleiotropic effects (what would RNA-seq transcriptomics of MTLL14-depleted cells look like). One can easily assume that many mRNAs are affected in experiments 7 to 9. This work, while interesting and promising, remains preliminary. If anything, instead of only profiling Elmo1 mRNA, the authors should include a panel to make a point about specificity (although unbiased approaches would be more powerful).

7. The study is highly focused on Elmo1. The authors should co-monitor the levels of Elmo2 and Elmo3 and some Dock proteins. This would give a better picture of the mechanisms at play.

MINOR COMMENTS

1. The introduction is incomplete regarding Elmo proteins biological functions. The authors should be more rigorous in citing these papers.

2. A potentially major issue, but I was not sure, is whether cell proliferation might be affected when Elmo1 is depleted. Did the authors verify this? In vitro, this is trivial, but could the bioluminescence data presented as "in vivo" be a mix of a reflection of migration and growth?

3. How was the Elmo1 antibody validated prior to use on human patient samples? How are the readers supposed to assess the validity of this experiment without any control or citation confirming the robustness of this antibody for human sample studies?

4. The discussion is very long and highly focused on repeating the main findings. This should be significantly shortened and used as an opportunity to discuss the data in the context of the field and how Elmo1 can be used as a therapeutic target.

Reviewer #2 (Remarks to the Author):

Overall, the writing was very clear and structured in an organized manner that allowed the reader to follow their thought process and experimental design. I also appreciate the representative images alongside the quantitative data they provided.

Some comments

- The staining in figure 5A should be accompanied with quantifiable data to demonstrate this phenomenon is not just unique to those two fields they found. I think providing a mean fluorescence intensity over multiple cells in both conditions could help demonstrate this
- The abstract does not contain method and results and it is just like an introduction.
- In line 91-93: As both the major source of osteoblasts and the critical regulator of immune cells, MSC play an important role as the bridge connecting bone metabolism and immune homeostasis - we need more references, since osteoblastic differentiation of mesenchymal stem cells is not approved in vivo.
- In line 492-495: In some (WHICH ONES?) assays, 2×10^5 macrophages differentiated from CD14+ monocytes isolated from peripheral blood samples were suspended in 600 μ l DMEM and then seeded in the lower chambers without or with 0.5 μ g/ml anti-TNF- α , anti-IL-17 or anti-IL-23 neutralizing antibody (R&D Systems.) also in line 133-135 Conversely, migration assays showed that the numbers of stained migratory AS-MSC were much greater than those of HC-MSC when cocultured with macrophages.) The authors mention the effect of the macrophages but the results are not included. The analysis has to be divided to 4 groups instead of two (AS- MSC, HC- MSC, MQ+AS- MSC, MQ+HC- MSC)

- Line 494: (0.5 µg/ml anti-TNF-α, anti-IL-17 or 495 anti-IL-23 neutralizing antibody) it is not correct way to use the same level of neutralizing antibody for different antibody, because the normal amount of these three are not the same.
- Supplemental Tables 5 not provided.
- Figure 8H didn't referenced in the text
- Conclusion is not provided
- Because the authors argued about high concentration of TNF which affects the directional migration of MSC and also, they work on AS patients sample. It proposed to measure the concentration of TNF in protein and mRNA level and other inflammatory cytokines like IL-17 and IL-23 systemically and locally in these samples and compare the results to be sure that AS patients have high concentration of TNF which was mention as the main key for MSC migration in this study.
- Blocking IL-17 or IL-23 no impact on migration. Why then is anti-IL17 mAb effective in AS?
- Why does only 100 ng/ml of TNF effect differential migration of MSCs AS vs HC? Is there no dose response? Is this a physiological level of TNF
- What does migration mean with respect to AS pathogenesis?
- How many AS patients and HC were used in the transcriptome data (Fig 3)?
- Is there a different expression of TNFR1 and TNFR2 on the MSCs in AS vs HC?
- If AS MSC enhanced osteoblastogenesis is mediated by BMP2 overexpression, how does that relate to ELMO?
- How does the ELMO effect on MSC migration compare with the IL-22 effect?
- Fig 5: is this convincingly demonstrated to be an enthesis? Where is the H&E?
- TNF inhibits MSC differentiation by m6A effects on miRNA...but seen also in SLE and RA: so not specific for AS?

Reviewer #3 (Remarks to the Author):

In this manuscript, Xie et al proposed ELMO1 as a novel potential therapeutic target in ankylosing spondylitis (AS) and demonstrated the significance of m6A modification in controlling the ELMO1 expression level in AS-MSc upon TNF-α stimulation. Overall, the evidence is quite solid and the manuscript was well written, with results being well organized and displayed. Nonetheless, some concerns need to be addressed before the manuscript is suitable for publish. I have the following suggestions to further strengthen this paper:

Major points:

1. The authors proposed that only high concentration of TNF-α (100 ng/mL) could enhance the migration ability of AS-MSc. Does this concentration equal to that found in AS patients? How do authors explain the phenomenon that HC-MSc and AS-MSc displayed similar migration ability with TNF-α treatment at lower concentrations (e.g., 10 and 50 ng/mL)?
2. Did authors check the migratory capacity (as claimed) of HC-MSc or AS-MSc when co-transplanted with macrophages into nude mice? This might mimic the real TNF-α level in vivo.
3. The authors used 3 siRNAs for testing and selected one with the highest knockdown efficiency to construct shRNA for loss-of-function studies for both ELMO1 and METTL14; however, this could not rule out the non-specific targeting. One more shRNA for each gene is suggested to make the data more convincing.
4. The authors proved the regulatory role of TNF-α in ELMO1 expression (Figure 3) and proposed METTL14 as a key regulator in manipulating the m6A modification in ELMO1 3' UTR (Figure 7,8 and 9), yet the connection between TNF-α and METTL14 was neglected. To justify the feedback regulation between TNF-α and AS-MSc, the mechanism on how TNF-α regulates METTL14 and how the coordination affects AS-MSc migration require further exploration.
5. The effect of m6A modification on mRNA transcripts is mediated by specific m6A readers. Did authors check whether the known m6A readers are responsible for ELMO1 mRNA degeneration after TNF-α stimulation?
6. Please provide the Co-IP results as mentioned in line 200 (supplementary figure 3).
7. Can authors provide results of METTL14 expression at the protein level in METTL14 knockdown or overexpression experiments (Supplementary 4)?

8. The assay using nude mice for transplanting MSC seems to reflect cell proliferation rather than the directional migration of transplanted cells. Could the authors explain a bit more?
9. The figure legends should provide description of related experiments rather than discussion of the results. More importantly, a legend should be understandable without the need to consult the text. It is strongly encouraged that the authors add necessary details in figure legends. For example, what do different colors in figure 3H stand for? What do specific abbreviations indicate, such as "W/O, TNFi"? In addition, please add the description of error bars in the corresponding figure legends.

Minor points:

1. The introduction part (Lines 122-129) just simply repeated the sentences in the Abstract (lines 66-73). Please revise to avoid that.
2. The sub title in Page 11 is inappropriate. METTL14 will not lead to low m6A modification, please correct.
3. Line 236, "we constructed Av-ELMO1 for SKG mouse treatment." Can authors provide experimental evidence that the expression level of ELMO1 in SKG mice treated with Av-ELMO1 is indeed reduced?
4. Line 296, the statement perhaps should be rephrased to "CLIP-PCR results showed that ELMO1 was enriched in the METTL14 group than in the IgG group."
5. It's necessary to add scale bars in corresponding figure, including but not limited to figure 5E, 5G.
6. Please annotate the sample name in Figure 5G.
7. Figure 6, panels F to I could be moved to supplementary figures.
8. Figure 9, panels D to F should be merged into one.
9. When citing references, it is highly recommended to cite original research articles rather than review articles.

Reviewer #4 (Remarks to the Author):

The manuscript by Xie et al. entitled "TNF- α -mediated m6A modification of ELMO1 triggers directional migration of mesenchymal stem cell in ankylosing spondylitis" aims at investigating the role of m6A modification of ELMO1 by TNF- α in migration of MSC. TNF α leads to increase migration of MSC through increase of ELMO1 expression. Silencing of ELMO1 leads to reduction of MSC migration as well as arthritis symptoms and overexpression of ELMO1 leads to increase of HC MSC migration. m6A modification of ELMO1 is regulated by METTL14 and it is decreased in high concentration of TNF α -treated MSC.

This is an interesting and timely manuscript. The experiments have been carefully carried out and most of the conclusions drawn are justified. There are no ethical concerns arising. The authors' claims are mostly convincing. On that account, I have only some minor suggestions below.

1. When comparing more than three groups, usually we use one way ANOVA or Kruskal Wallis test. However the statistical information is lacking in method section. The number of samples are relatively low (N=6 to 9), I think the authors should use Kruskal Wallis test. In addition, the significance should be presented by comparing to each group (figure 4,5,7,8,9). As presented in method section of statistics, why the author used mean with SD? I think using nonparametric statistical method is suitable for current study, because number of each group was small.
2. The results are commented in the most of figure legends, whereas there should be no comment of results in figure legends.
3. In figure 2B, 4C, H, 7H, and 8H, injected MSC migrated less or more into TNF α injected site. In case of less migrated mice, it seems that MSC are staying at first injected site. Is there any data available that supports the authors' conclusions?
4. In figure 5, authors showed that silencing of ELMO1 has protective effect in arthritis. However, in figure 5 legend (line 954-957), the explanation is opposite. Please revise the sentence.
5. In figure 5, Did the arthritis score and incidence were significantly differ between AV-ELMO1 group vs PBS or AV-NC. The figure lacks the significance. Furthermore, the MSC is the progenitor

of osteoblast, and inhibiting MSC in SKG mice could ameliorate syndesmophyte formation in vivo SpA model. Did the author tried to quantify the syndesmophyte formation in micro CT or histology?

6. Furthermore, the incidence rate reach similar between PBS, Av-NC, and Av-ELMO1 group (figure 5C). Do authors has hypothesis why the effect of ELMO1 inhibition disappeared (or weakened) after 7 weeks ?

Response To Reviewers Letter

Dear Reviewers:

Thank you for the constructive comments concerning our manuscript entitled “TNF- α -mediated m⁶A modification of ELMO1 triggers directional migration of mesenchymal stem cell in ankylosing spondylitis” (Manuscript ID: NCOMMS-20-35339). These comments were valuable and very helpful for improving our manuscript to better demonstrate the important significance of our research. We have carefully reviewed all comments and completed point-by-point revisions. The responses to the comments in this letter and the revised portions of the manuscript are marked in colors. We appreciate the work of Reviewers and hope that the revisions will meet with approval. We will be glad to respond to any further comments that you may have.

Yours sincerely,

Huiyong Shen

Department of Orthopedics, The Eighth Affiliated Hospital, Sun Yat-sen University,
3025# Shennan Road, Shenzhen, 518000, P.R. China.

Tel: +86 139 2227 6368

Fax: +86 20 8133 2612

Email: shenhuiy@mail.sysu.edu.cn

Responses to Reviewer 1

In this manuscript, the authors aim to define the mechanisms whereby TNFa could be
contributing to ankylosing spondylitis (AS). They define that high dose TNFa
increases migration of AS-MSC in cell and graft models in vivo. By conducting
transcriptomics experiments, they identify a large network of modulated genes. They
focus on Elmo1 since it is integrating many pro-migratory cytokine receptor pathways.
Depletion of Elmo1 decreases migration in cells as well as in vivo in graft assay
(TNFa-induced). A central experiment is the in vivo depletion of Elmo by systematic
siRNA injection in a model of SGK mice that appears to decrease arthritis. Next, the
authors investigated the mechanisms of Elmo1 over expression in AS-MSC cells. The
identified the m6A modifier MTTL14. Depletion of MTTL14 increases Elmo1
expression while its over-expression decreased the stability of the Elmo mRNA. They
identified a region in the Elmo UTR that is sensitive to MTTL14 activity.

Overall, the data presented is technically solid. This is the biggest strength of he
manuscript. However, it fails to bring a major mechanistic contribution given that
(1)-Elmo proteins have already been tightly linked to increased migration, (2)
knockout of Elmo has been reported to dampen the features of arthritis and (3) the
MTTL14 story is poorly connected to the rest of the story. The cell biology presented
in this paper is also very preliminary and over-interpreted.

We appreciate the constructive comments by Reviewer 1. Based on these comments,
we revised the manuscript as follows. The responses to the comments by Reviewer 1
in this letter and the revised portions of the manuscript are marked in red.

MAJOR COMMENTS.

1. The patient data is barely discussed in the manuscript, but instead in the Methods
section. This is quite unusual and not appropriate for such a manuscript where this
info should be clearly presented in the main text.

**Response: Due to the limitation of the displayed items in the manuscript, the patient**
**information, including age, disease duration, inflammatory biomarkers and disease**
**activity score, are shown in Supplemental Tables 1 and 4. According to this comment,**
**we added the patient information in the Results and Discussion sections (Lines**
**134-136, 238-239, and 345-346). If the reviewer considers these descriptions**
**insufficient, we could include the supplemental tables in the main text of the**
**manuscript.**

2. The authors claim that Elmo1 promotes, and localizes to, filopodia in AS-MSK.
This is very poorly conducted in the manuscript. First the Elmo-Dock-Rac signaling
axis would be more consistent with lamellipodia formation. This is not discussed.
Also, naming cellular extensions as “filopodia” is premature without proper staining
for markers of these structures. This data should all be removed if not analyzed
correctly as it fails to provide any new insights and is largely overinterpreted.

**Response: Thank you for this valuable comment. According to the comment by**
**reviewer 1, we chose to remove these data from our manuscript. Additionally, the**
**description of the ELMO1 immunofluorescence results was revised (Lines 193-195**
**and 997-999).**

3. Elmo1 is already tightly connected to cell migration. In this manuscript, the authors

fail to provide any real insights on how it might function other than promote Rac
activation. Even the Rac connections is not formally demonstrated. One would
assume Elmo1 signals via Dock proteins (1,4 and 5 expressed in their cells). This
should at a minimum be validated with siRNAs. The authors provide proteomics data
on Elmo1 immunoprecipitates in +/- TNFa to attempt to make news connections (I
assume), but this only reveals known interactor. This data should clearly be removed
from this manuscript. In a high-profile journal, if the data is not used or dissected
correctly, it is not useful. One idea would have been to begin connecting a novel
chemokine receptor upstream of Elmo. The authors mention that there could be
redundancy among those, but it has not been tested really. This may in fact uncover
yet another clinical intervention point.

**Response: Thank you very much for this insightful comment.**

**First, several experiments were performed in response to this comment. ① Protein**
**was extracted from the TNF- α treated HC-MSC and AS-MSC, and Co-IP assays were**
**performed using an anti-ELMO1 antibody and an IgG control. Then, Western blot**
**analyses were performed to detect the level of ELMO1, DOCK1, DOCK4 and**
**DOCK5. Consistent with the Co-IP and LC-MS/MS results, the findings showed that**
**ELMO1 bound DOCK1, DOCK4 and DOCK5 in both HC-MSC and AS-MSC**
**(Revision Figure 1A and B). ② siRNAs for DOCK1, 4 and 5 were constructed and**
**verified. MSC were first transfected with OE-NC or OE-ELMO1, and the**
**OE-ELMO1 transfected MSC were further transfected with these siRNAs using**
**Lipofectamine RNAi MAX. The active Rac1 levels in all groups were detected after**

the TNF- α treatment. The results showed that ELMO overexpression in MSC
 significantly increased the active level of Rac1 following the TNF- α treatment, and
 this effect was counteracted only by inhibiting DOCK1 expression in MSC (Revision
 Figure 1C). Inhibiting DOCK4 or DOCK5 did not affect the active Rac1 level when
 over-expressing ELMO1 in MSC. These results suggest that ELMO1 binds DOCK1
 and promotes Rac1 activation in TNF- α treated MSC. These results were
 supplemented in Supplemental Figure 4. Please see Lines 208-210.

**Revision Figure 1**

Previous studies have demonstrated that ELMO1 bound DOCK1 and formed an
 ELMO1/DOCK1 complex. This complex acts as a guanine nucleotide exchange
 factor that activates Rac1 in endothelial cells¹. This mode of action of
 ELMO1/DOCK1 was proven in other cells^{2,3}. Moreover, Lu's study revealed that the
 C-terminal polyproline region of ELMO1 (707 to 714 amino acids) interacts with the

SH3 domain of Dock1 in 293T cells⁴. In this study, for the first time, we investigated
the ELMO1 binding protein in MSC and found that ELMO1 mainly bound the DOCK
family proteins (1, 4 and 5) in MSC as in other cells. Additionally, using siRNAs, we
determined that ELMO1 activates Rac1 mainly by binding DOCK1 in TNF- α treated
MSC. Unquestionably, it is of great importance to reveal the interaction site between
ELMO1 and DOCK1 and the mechanism by which they activate Rac1 in MSC. Due
to the emphasis we placed on the TNF- α /METTL14/ELMO1 axis in AS pathogenesis
and the length limitation of the submission, we will continue our further studies
concerning this issue in the future.

In this study, we performed Co-IP and LC-MS/MS assays in TNF- α treated MSC.
Instead of revealing new interaction relationships of ELMO1 in MSC, the aim of this
assays were to ① investigate the binding protein of ELMO1 in MSC and determine
whether these interactions were similar to those in other cells, which have never been
previously studied, and ② determine whether differences exist in ELMO1 binding
proteins between TNF- α treated HC-MSC and AS-MSC. Based on our results, we
demonstrated that the binding proteins of ELMO1 in MSC were similar to those in
other cells, and few differences were observed between HC-MSC and AS-MSC. As
reviewer 1 indicated, only the reported protein-protein interaction relationship could
be identified in our study using the STRING dataset. However, we continued our PPI
analysis based on Supplemental Tables 5 and 6, but no special chemokine receptors
were found in our Co-IP and LC-MS/MS results. Apparently, it is of great significance
to study upstream of ELMO1 in further research. However, in our study, we aim to

investigate the mechanism underlying the enhanced migration ability of AS-MSK and
the pathogenesis of AS. Therefore, we consider retaining this part of the data, and we
will continue our further studies aiming to identify the upstream proteins of ELMO1.
If the reviewer insists on removing these data, we will delete these results according
to the comments in the next revision.

4. I have a major issue with the mouse model used as a pre-clinical model. First of all,
it is poorly presented in the manuscript, even misleading to the reader. The SGK mice
were treated to Curdlan to induce the features of AS (as done by others in the
literature). This is not mentioned at all but in the methods. Most importantly, the
authors did not analyze the expression of Elmo in this model. Is Elmo induced? This
is critical since if it is Elmo-independent, then the knockdown of Elmo1 by siRNA
may be of pleiotropic origin rather than really targeting the vicious cycle studied here
(TNF α , Elmo1 expression). The model must be validated as a model useful to assess
the contribution of Elmo1. Also, why didn't the author cross Elmo1 $-/-$ mice with their
SGK mouse model? This may have been a cleaner genetic experiment. Finally, it is
never discussed clearly how Elmo1 might be targeted in vivo. One guess would be
that the Dock protein would be the ideal target as inhibitors have been developed of
the years (Elmo being a scaffold).

Response: Thank you for this helpful comment.

① We apologize for our misleading presentation of the *in vivo* experiments using the
SGK mice. According to this comment, we supplemented the induction and features
of the SGK mice in the results and discussion sections. Please see lines 242-243 and

389-392.

② In our preliminary experiment, we isolated MSC from SKG mice at 0, 4 and 8
149 week after the curdlan induction and then detected their ELMO1 expression. The
150 results showed that ELMO1 expression in MSC from the SKG mice was increased
from 0 to 8 weeks after the curdlan induction in the PBS group and Av-NC group.
ELMO1 expression in the MSC from the SKG mice was significantly inhibited in the
Av-ELMO1 group at 4 and 8 weeks compared to that in the other two groups
(Revision Figure 2A). Additionally, the results of the immunofluorescence assays
showed that ELMO1 expression in MSC was increased after induction at 8 weeks and
that ELMO1 expression in MSC was lower than that in the other two groups at 8
157 weeks (Revision Figure 2B). These results showed that ELMO1 expression was
158 induced in the SKG mice after the curdlan induction and that Av-ELMO1 could target
MSC and inhibit ELMO1 expression. Consistent with other previous studies⁵⁻⁷, these
results confirm that SKG mice serve as a useful AS model for the assessment of the
contribution of ELMO1 in AS.

Revision Figure 2

③ Previous studies have demonstrated that ELMO1 is widely expressed in various
types of cells. In addition, Arandjelovic's study showed that ELMO1 in neutrophils
promoted inflammatory arthritis². According to our preliminary experiment and
results shown in the manuscript, ELMO1 expression in MSC was abnormally
increased *in vitro* and *vivo* (enthesitis tissue), and ELMO1 expression in MSC from
SKG mice was increased after the curdlan induction and decreased with the
Av-ELMO1 injection. Moreover, inhibiting ELMO1 expression by Av-ELMO1
significantly improved the local inflammation and histological changes in the SKG
mice. As we indicated in the manuscript (Lines 394-397), we could not conclude
whether Av-ELMO1 affected cells other than MSC in this study. However,
Av-ELMO1 decreased ELMO1 expression in MSC from the SKG mice and then
markedly reduced the numbers of migratory CD105+ MSC and local inflammation
(Figure 5G), confirming the critical role of ELMO1 in MSC in the pathogenesis of
AS.

④ We completely agree with the reviewer's comment regarding the construction of
KO or CKO mice with an SKG genetic background. However, due to the COVID-19
effect on animal imports and the large requirement of time and expenditure, we are
unable to currently conduct this experiment. However, we aim to perform this
research and present our results as soon as possible.

⑤ The reviewer's comment regarding the ELMO1/DOCK axis is insightful and
constructive. Recently,

4-[3'-(2-chlorophenyl)-2'-propen-1'-ylidene]-1-phenyl-3,5-pyrazolidinedione (CPYPP)

was designed as a small molecule inhibitor of DOCK2 and showed great ability in
releasing inflammation^{8,9}. This inhibitor has been used to antagonize ELMO1/DOCK
Rac1 GEF activity¹⁰. According to our study, the TNF- α /METTL14/ELMO1 axis in
MSC plays an important role in AS pathogenesis. Theoretically, breaking a node in
this axis could have a therapeutic effect on AS, and targeting ELMO1/DOCK using
CPYPP can be preferentially studied because of its mature commercialization. This
information was added to the discussion section. Please see lines 483-490.

5. The MTTL14 story is potentially very interesting but appears randomly in the study.
Did the authors find any evidence in their RNA-seq data that MTTL14 expression is
decreased in AS-MSC treated with TNF? I fail to see any “biological process” or
“machinery” in the presented data that would suggest it is the case. What about in
the patient samples?

Response: Thank you for this helpful comment. We investigated METTL14
expression in the RNA sequencing data when conducting our study. However, no
difference in METTL14 was observed between HC-MSC and AS-MSC in the RNA
sequencing data. This result was confirmed by qRT-PCR (Revision Figure 3A).
Therefore, we suggest that TNF- α may directly regulate METTL14 expression at the
protein level, but not the RNA level, in MSC.

Previous studies have proven that protein degradation is one of the most common and
important posttranslational modification patterns¹¹. This posttranslational
modification pattern occurs mainly through the autophagy-lysosome pathway and
ubiquitin-proteasome pathway^{12,13}. Therefore, we used MG132, which is a

proteasome inhibitor, and chloroquine (CQ), which is an autophagy inhibitor, to
investigate the role of these two pathways in TNF- α mediated METTL14 expression
in MSC. The results showed that the TNF- α treatment decreased the METTL14
protein level in MSC, and adding MG132 significantly antagonized this effect and
up-regulated METTL14 expression. Additionally, CQ did not affect METTL14
expression in the TNF- α treated MSC (Revision Figure 3B). These results indicate
that TNF- α decreased the METTL14 protein level in MSC through a posttranslational
modification pattern involving the ubiquitin-proteasome pathway.

Jian's study showed that TNF- α stimulation increased METTL14 expression at the
mRNA and protein levels in endothelial cells¹⁴. This discrepancy may result from the
different cells investigated in the studies. However, how TNF- α affects METTL14
expression was not studied in their research. A previous study demonstrated that the
ubiquitin-proteasome system could be partially damaged in AS, contributing to the
development of inflammation¹⁵. Combined with our supplemental results, we suggest
that the decrease in METTL14 induced by TNF- α through a posttranslational
modification pattern may result from a dysfunction in the ubiquitin-proteasome
system in AS-MSC after TNF- α treatment, which requires further investigation.

In addition, we detected METTL14 expression in MSC in human tissues and
confirmed that METTL14 expression in CD105⁺ MSC in the AS enthesis samples was
lower than that in the control enthesis samples (Revision Figure 3C). The results of
METTL14 expression in MSC in human tissues were added in the new Figure 6F.

Please see lines 267-269 and 428-430.

Revision Figure 3

6. While the METTL14 data is per se clean and convincing, the authors completely
 overlook the fact that interfering with METTL14 may have pleiotropic effects (what
 would RNA-seq transcriptomics of METTL14-depleted cells look like). One can easily
 assume that many mRNAs are affected in experiments 7 to 9. This work, while
 interesting and promising, remains preliminary. If anything, instead of only profiling
 Elmo1 mRNA, the authors should include a panel to make a point about specificity
 (although unbiased approaches would be more powerful).

**Response: We appreciate this insightful comment.**

**First, we detected the RNA expression profiles of three MSC transfected with**
 **Lv-METTL14 and three matched MSC transfected with Lv-NC using RNA**
 **sequencing (Revision Figure 4A). The lentivirus transfection and RNA sequencing**
 **were performed according to the methods described in the manuscript. The data**

revealed 950 differentially expressed mRNAs between the Lv-NC group and Lv-M14
group, including 389 up-regulated mRNAs and 561 down-regulated mRNAs
(Revision Figure 4B). GO and KEGG analyses were performed to investigate the
function and enriched pathways of these differentially expressed mRNAs. The results
of the KEGG analysis showed that the enriched pathways were related to cancer (such
as pathways in cancer and proteoglycans in cancer) and several signal transduction
pathways (such as the NOD-like pathway, Rap1 pathway, PI3K-Akt pathway, TNF
pathway, and Chemokine pathway) (Revision Figure 4C). Notably, 15 differentially
expressed mRNAs were involved in the Chemokine signaling pathway, and ELMO1
was an up-regulated mRNA with the highest fold change (+4.18). Additionally, the
GO analysis in biological process determined that these differentially expressed
mRNAs were related to many different types of physiological functions, including
immunity, angiogenesis, migration and differentiation (Revision Figure 4F).

Second, the MSC transfected with Lv-ELMO1 were further divided into three groups
(control group, Lv-NC group and Lv-M14 group) and transfected without or with
Lv-NC (or Lv-M14). The migration ability of the three groups was detected using
migration assays in a Transwell system and wound healing assays according to the
methods described in the manuscript. However, no differences in migration ability
were observed among the three groups (Revision Figure 4G and H).

N⁶-methyladenosine (m⁶A) is the most abundant internal mRNA modification
regulated by methyltransferases and demethylases¹⁶. Therefore, undoubtedly, m⁶A
modification can affect many different functions of different cells. As a critical m⁶A

methyltransferase, we agree with the comment by Reviewer 1 that METTL14 may
have pleotropic effects in MSC. Through our supplemental experiments, as shown in
Revision Figure 4, we determined that 950 mRNAs were affected after inhibiting
METTL14 expression in MSC. Additionally, in addition to cell migration, the
bioinformatics analysis suggested that these differentially expressed mRNAs were
enriched in various signal pathways and related to various aspects of biological
processes in MSC. These results support the speculation that METTL14 has
pleotropic effects in cells, including MSC. However, functional experiments should
be further performed to confirm the effect of METTL14 on the detailed functions of
MSC. To the best of our knowledge, the effect of METTL14 on MSC functions has
not been investigated, and our research is the first to demonstrate that METTL14
regulates the migration potential of MSC.

Although METTL14 may have pleotropic effects and regulate many genes, we
suggest that ELMO1 is critical for METTL14-mediated MSC migration due to the
following reasons. ① In the METTL14 depleted RNA sequencing, cell migration
and the chemokine signaling pathway were enriched in the GO and KEGG analyses.
Moreover, ELMO1 was an up-regulated mRNA with the highest fold change (4.18).
② When ELMO1 expression was inhibited, the decreased METTL14 expression had
lower effect on MSC migration. ③ ELMO1 was abnormally expressed in AS and
correlated with the enhanced migration ability of AS-MSC. Inhibiting ELMO1
expression in AS-MSC rectified enhanced migration ability. Therefore, we implicate
the specificity of the METTL14/ELMO1 axis in (AS-)MSC migration regulation.

Revision Figure 4

7. The study is highly focused on Elmo1. The authors should co-monitor the levels of
 Elmo2 and Elmo3 and some Dock proteins. This would give a better picture of the
 mechanisms at play.

**Response:** Thank you for this constructive comment.

When scanning the RNA sequencing data, no differences in ELMO2 or 3 and DOCK1,
 2, 4, 5, or 8 were observed between HC-MSC and AS-MSC. Then, we further
 detected the protein levels of these proteins. The results show that ELMO3 and
 DOCK8 were less expressed in MSC. Additionally, the expression levels of ELMO2,
 DOCK1, DOCK2, DOCK4 and DOCK5 were equal between HC-MSC and AS-MSC

with the TNF- α treatment (Revision Figure 5). These results were added in the
manuscript as a new Supplemental Figure 3. Please see lines 193-198.

Revision Figure 5

MINOR COMMENTS

1. The introduction is incomplete regarding Elmo proteins biological functions. The
authors should be more rigorous in citing these papers.

Response: We apologize for our unclear statement regarding ELMO1. Due to the
length limitation of the manuscript, we simply introduced the functions of ELMO1 in
the Introduction. We revised the introduction related to ELMO1 and the references.

Please see lines 105-112.

2. A potentially major issue, but I was not sure, is whether cell proliferation might be
affected when Elmo1 is depleted. Did the authors verify this? In vitro, this is trivial,
but could the bioluminescence data presented as “in vivo” be a mix of a reflection of
migration and growth?

Response: We appreciate the reviewer’s insightful comment. Actually, we detected the
proliferation ability of MSC transfected with Lv-ELMO1 or OE-ELMO1. As shown
by the CCK-8 results, no differences in the proliferative rate were observed between

the MSC transfected with Lv-ELMO1 (or OE-ELMO1) and their NC group (Revision
Figure 6). These results were supplemented in the main text and Supplemental Figure
5. Please see lines 215-217 and 226-228.

Revision Figure 6

3. How was the Elmo1 antibody validate prior to use on human patient samples? How
are the readers supposed to assess the validity of this experiment without any control
or citation confirming the robustness of this antibody for human sample studies?

Response: Thank you for this helpful comment. In each tissue immunohistochemical
and immunofluorescence staining assay, we used corresponding antibodies as isotype
controls to insure the authenticity of the results. According to this comment, we listed
the sources and identifiers of the reagents and antibodies used in our experiments to
allow the reader to assess the validity. Please see new Supplemental Table 7.

4. The discussion is very long and highly focused on repeating the main findings. This
should be significantly shortened and used as an opportunity to discuss the data in the
context of the field and how Elmo1 can be used as a therapeutic target.

Response: Thank you for this valuable comment. The discussion section was
shortened, and several new discussion issues were added. The therapeutic effect of
ELMO1 was discussed in the discussion section. Please see lines 439-451, 483-490.

Responses to Reviewer 2

Overall, the writing was very clear and structured in an organized matter that allowed
the reader to follow their thought process and experimental design. I also appreciate
the representative images alongside the quantitative data they provided.

We appreciate the positive comments by Reviewer 2 in the opening sentence. Based
on these helpful comments, we revised the manuscript as follows. The responses to
the comments by Reviewer 2 in this letter and the revised portions in the manuscript
are marked in blue.

Some comments

1. The staining in figure 5A should be accompanied with quantifiable data to
demonstrate this phenomenon is not just unique to those two fields they found. I think
providing a mean fluorescence intensity over multiple cells in both conditions could
help demonstrate this

Response: Thank you for this constructive comment. We analyzed the mean
fluorescence intensity of ELMO1 to quantify its expression level using ImagePro 6.0
software. The results showed that ELMO1 expression in the AS patient tissues was
higher than that in the non-AS patient tissues. This information was supplemented in
Figure 5A, and the Results, Methods and Legends were also revised. Please see lines
239-242.

2. The abstract does not contain method and results and it is just like an introduction.

Response: Because the submission guidelines regarding the abstract state “Provide a
general introduction to the topic and a brief nontechnical summary of your main

results and their implication” and the word limitation (150 words), we exerted our
best effort to condense the text of the abstract to the greatest extent possible.
Additionally, the abstract of *Nature Communications* does not include four parts
(introduction, methods, results and conclusion) as other journals.

3. In line 91-93: As both the major source of osteoblasts and the critical regulator of
immune cells, MSC play an important role as the bridge connecting bone metabolism
and immune homeostasis - we need more references, since osteoblastic differentiation
of mesenchymal stem cells is not approved *in vivo*.

Response: Thank you for your elaborative comment. MSC were first identified in the
1970s as a type of multipotent stem cells *in vivo*¹⁷. In recent years, many studies
focusing on MSC have been conducted, and the following three main aspects of MSC
function were identified: ① MSC support hematopoietic stem cells in the bone
marrow as niche cells¹⁸⁻²⁰; ② MSC exhibit their immunoregulatory capacity by
affecting various immune cells, such as T cells^{21,22}, B cells^{23,24} and macrophage;^{25,26}
and ③ MSC possess trilineage differentiation abilities, including osteogenic,
chondrogenic and adipogenic differentiation²⁷. Specifically, MSC are the major
origin of osteoblasts and play an important role in bone metabolism²⁸. Although some
differences may still exist, some researchers consider the terms skeletal stem cells and
MSC interchangeable, highlighting the critical role of MSC in bone development^{29,30}.
Using CKO mice to conduct *in vivo* experiments, Morrison’s study demonstrated that
LepR-expressing MSC are a major source of bone in adult bone marrow³¹. Moreover,
Prof. Scadden revealed that osteoblastic cells should be replenished from

bone-marrow-derived MSC through genetic pulse-chase experiments *in vivo*³².
Several other *in vivo* studies further confirmed this conclusion³³⁻³⁵. Additionally, we
added references according to this comment. Please see lines 91-93.

4. In line 492-495: In some (WHICH ONES?) assays, 2×10^5 macrophages
differentiated from CD14+ monocytes isolated from peripheral blood samples were
suspended in 600 μ l DMEM and then seeded in the lower chambers without or with
0.5 μ g/ml anti-TNF- α , anti-IL-17 or anti-IL-23 neutralizing antibody (R&D
Systems.) also in line 133-135 Conversely, migration assays showed that the
numbers of stained migratory AS-MSC were much greater than those of HC-MSC
when cocultured with macrophages.) The authors mention the effect of the
macrophages but the results are not included. The analysis has to be divided to 4
groups instead of two (AS- MSC,HC-MSC,MQ+AS-MSC,MQ+HC-MSC)

Response: We apologize for our misleading description and presentation of the data.

① “In some assays” on line 492 indicates the assays shown in Figure 1A and
Supplemental Figure 1. We corrected our unclear description in the manuscript. Please
see lines 523-524.

② The sentences on lines 133-135 were also misleading, and we made revisions. The
results shown in Figure 1A and Supplemental Figure 1 reflect the effect of
macrophages on HC-MSC and AS-MSC. Please see lines 136-143. All groups shown
in Figure 1A and Supplemental Figure 1 were cultured with macrophages using a
Transwell system. We also clarified this issue in the new figure. Please see Figure 1A
and Supplemental Figure 1.

5. Line 494: (0.5 μ g/ml anti-TNF- α , anti-IL-17 or 495 anti-IL-23 neutralizing
antibody) it is not correct way to use the same level of neutralizing antibody for
different antibody, because the normal amount of these three are not the same.

Response: We appreciate the elaborative review of our manuscript. We added the
catalog number of the three antibodies and the dosage of the anti-IL-17 and anti-IL-23
neutralizing antibodies in the manuscript. Please see lines 526-527.

① The anti-TNF- α neutralizing antibody (MAB610) was purchased from R&D
System Company. According to the protocol, the neutralization dose (ND50) is
typically 0.01-0.04 μ g/mL in the presence of actinomycin D and 0.25 ng/mL
Recombinant Human TNF- α . Using an ELISA assay, our preliminary experiment
showed that the TNF- α level in the Transwell system was approximately 1000 pg/ml.
Therefore, the dosage of the anti-TNF- α neutralizing antibody we used in the
experiment was 0.5 μ g/mL.

② The anti-IL-17 neutralizing antibody (AF-317) was also purchased from R&D
System Company. The ND50 is typically 0.02-0.12 μ g/mL in the presence of 15
417 ng/mL Recombinant Human IL-17. The IL-17 level in the Transwell system was
418 rather low (approximately 10 pg/ml) in our preliminary experiment. The dosage of the
419 anti-IL-17 neutralizing antibody we used in the experiment was 0.05 μ g/mL.

③ The ND50 of the anti-IL-23 neutralizing antibody (AF-1716, R&D System
Company) is typically 0.2-0.8 μ g/mL in the presence of 0.75 ng/mL Recombinant
Human IL-23. The IL-23 level in the Transwell system was approximately 200 pg/ml
in our preliminary experiment. Therefore, the dosage of the anti-IL-3 neutralizing

antibody we used in the experiment was 0.5 µg/mL.

6. Supplemental Tables 5 not provided.

Response: In the previous submission, we uploaded Supplemental Tables 5 and 6
showing the Co-IP and LC-MS/MS data as two Excel files. This format may affect the
display in the review file. Therefore, we changed the Excel files to Word files. Please
see the new Supplemental Tables 5 and 6.

7. Figure 8H didn't referenced in the text.

Response: We apologize for our carelessness. Figure 8H was supplemented in the text.
Please see lines 297-300.

8. Conclusion is not provided

Response: According to the submission guidelines of *Nature Communications*, a
conclusion is not included in the main text. Therefore, the last paragraph in the
discussion was the conclusion of our manuscript. Please see lines 491-496.

9. Because the authors argued about high concentration of TNF which affects the
directional migration of MSC and also, they work on AS patients sample. It proposed
to measure the concentration of TNF in protein and mRNA level and other
inflammatory cytokines like IL-17 and IL-23 systemically and locally in these
samples and compare the results to be sure that AS patients have high concentration of
TNF which was mention as the main key for MSC migration in this study.

Response: Thank you very this helpful comment.

According to the comment, we detected the TNF-α level in the serum of AS patients
by ELISAs and the TNF-α protein and mRNA levels in local tissue by qRT-PCR and

immunohistochemical assays, respectively. The results showed that the TNF- α level in
the serum of the AS patients was equal to that in the healthy controls (Revision Figure
7A). Additionally, both the mRNA level and the protein level of TNF- α in the local
tissue of the AS patients were significantly higher than those in the healthy controls
(Revision Figure 7B & C). These results indicate that TNF- α was elevated locally
rather than systemically.

Previous studies have systematically investigated the TNF- α level in AS patients, but
the results were controversial. Several studies determined that the TNF- α levels in the
serum of AS patients were higher than those in healthy controls^{36,37}. In contrast, some
other studies found that systemically, the TNF- α levels in the serum of AS patients
were similar to those of healthy control^{38,39}. According to our results, we confirmed
that the TNF- α level in the AS patients was elevated compared to that in the healthy
controls. However, this abnormal enhancement in TNF- α was local rather than
systemic in the pathological tissue of AS, which was critical for AS-MSC migration in
this study.

Due to the limitation of the manuscript length, this result is only shown in Revision
Figure 7. If the reviewer considers this result of great importance, we could add it to
the main text in the next revision. Additionally, since IL17/IL23 are not the major foci
of our manuscript, we did not further measure their levels in our study.

Revision Figure 7

10. Blocking IL-17 or IL-23 no impact on migration. Why then is anti-IL17 mAb
effective in AS?

Response: AS is a common type of rheumatic disease in which many pathological
mechanisms are involved. The abnormalities in T cells⁴⁰, macrophages⁴¹, and MSC
reported in our and other studies^{42,43} have been reported to contribute to AS. In recent
472 years, the IL-17/IL-23 axis was considered critical for the pathogenesis of AS⁴⁴, and
473 this pathway could be a therapeutic target for AS by affecting Th17 cells, γ/δ T cells
and mucosa-associated invariant T cells^{45,46}. Therefore, we suggest that an anti-IL17
mAb may not directly impact MSC through the mechanism in our manuscript while
exerting its therapeutic effect.

11. Why does only 100 ng/ml of TNF effect differential migration of MSCs AS vs HC?

Is there no dose response? Is this a physiological level of TNF

Response: Thank you for your helpful comment. We paid close attention to this issue
during our study. In this study, we found that the migratory ability of HC-MSC and
AS-MSC was enhanced as the TNF- α concentration increased from 10 to 100 ng/ml.
When treated with 100 ng/ml TNF- α , AS-MSC showed a stronger migration ability
than HC-MSC. This difference between AS-MSC and HC-MSC was also observed in

the 150 and 200 ng/ml TNF- α groups in our preliminary experiment. Actually, this
level (>100 ng/ml) is a pathological rather than physiological level of TNF- α *in vivo*.
MSC constitute a type of immunoregulatory cells *in vivo*⁴⁷, and TNF- α secreted by
other immune cells recruited MSC to a specific site to exert its functions^{48,49}. We
suggest that two distinct functions (pathological and physiological) of MSC and
TNF- α may exist in AS patients according to our results.

① In pathological sites, such as entheses in AS, inflammatory cells, especially
macrophages, widely infiltrate and express a large amount of TNF- α at a pathological
level^{50,51}. The pathological level of TNF- α (>100 ng/ml) leads to the enhanced
migration capacity of AS-MSC, resulting in disease development as we suggested in
the discussion (lines 353-360).

② In non-pathological lesion sites in AS patients, pathogenic factors, such as
infection or injury, induce normal inflammation to secrete physiological amounts of
TNF- α (much less than 100 ng/ml). This physiological level of TNF- α also recruits
moderate numbers of MSC to exert their function, but the migration ability of these
MSC is not abnormally enhanced.

Previous studies and our results (Revision Figure 7) demonstrated that TNF- α is
elevated locally rather than systemically in pathological tissue in AS³⁹. Therefore, the
different levels of TNF- α in AS patients *in vivo* may induce physiological or
pathological phenomena of MSC, and this result may explain the axial involvement
(the site of higher TNF- α in AS patients, such as spine entheses) in AS. We discussed
this issue in the discussion section (lines 353-360). Certainly, this issue should be

addressed in future studies.

12. What does migration mean with respect to AS pathogenesis?

Response: MSC are located in the bone marrow inside bone, but chronic
inflammation and ectopic ossification in AS occur outside bone, such as entheses [52].

However, how AS-MSC migrate from the bone marrow to the bone surface and
participate in AS development is unclear. Recently, microscopic holes connecting
entheses to the bone marrow were observed in cortical bone, and bone marrow tissues
were shown to spill into entheses through these holes, providing an anatomical basis
for MSC migration⁵³. Based on our results in the manuscript, we suggest that a higher
concentration of TNF- α in entheses recruits much more AS-MSC by enhancing their
migration ability. These AS-MSC migrate through microscopic holes to the bone
surface and contribute to the pathogenesis of AS. Previous research has indicated that
the abnormal migration ability of MSC was involved in the pathogenesis of several
diseases⁵⁴⁻⁵⁶. Therefore, these results reveal the importance of the cell migration
process, especially MSC migration, in AS pathogenesis.

13. How many AS patients and HC were used in the transcriptome data (Fig 3)?

Response: As indicated in the Results section, six HC-MSC and six AS-MSC were
used for the whole transcriptome sequencing. Please see lines 166-167. A heatmap of
the total sample of 24 is shown in Figure 3A.

14. Is there a different expression of TNFR1 and TNFR2 on the MSCs in AS vs HC?

Response: According to this helpful comment, we detected the protein levels of
TNFR1 and TNFR2 in HC-MSC and AS-MSC. The results showed no differences in

the expression levels of TNFR1 and TNFR2 between HC-MSC and AS-MSC
(Revision Figure 8).

Revision Figure 8

15. If AS MSC enhanced osteoblastogenesis is mediated by BMP2 overexpression,
how does that relate to ELMO?

Response: Thank you for this helpful comment.

Several experiments were performed to answer this question. ① Protein was
collected from MSC on days 0 to 21 of osteogenic differentiation, and ELMO1
expression was detected using a Western blot analysis. The results showed that as
MSC underwent osteogenesis, the ELMO1 expression levels remained almost
unchanged (Revision Figure 9A). ② ELMO1 was inhibited (Lv-ELMO1) or
over-expressed (OE-ELMO1) in MSC by lentivirus transfection, followed by Alizarin
Red S (ARS) and alkaline phosphatase (ALP) assays to determine the osteogenic
differentiation ability of MSC. However, as shown by the ARS and ALP assays, the
osteogenesis potential of MSC was unaffected by Lv-ELMO1 or OE-ELMO1
(Revision Figure 9B and C).

Our previous study demonstrated that a BMP/Noggin imbalance led to enhanced
osteogenic differentiation in AS-MSC⁴². According to our results above, ELMO1

expression was almost invariable during MSC osteogenic differentiation, and
 regulating ELMO1 expression did not alter the osteogenic differentiation ability of
 MSC. Therefore, we suggest that ELMO1 only participates in the dysfunction of the
 migration ability, but not osteogenic differentiation, of AS-MSC.

Revision Figure 9

16. How does the ELMO effect on MSC migration compare with the IL-22 effect?

Response: According to this comment, we compared the effect of ELMO1 and IL22

on MSC migration. As determined by the Transwell migration assays, more cells were

stained in the OE-ELMO1 group than the OE-NC and OE-NC+IL22 groups.

Additionally, the stained cells in the OE-NC+IL22 group were larger than those in the

OE-NC group (Revision Figure 10A & C). These results were confirmed and are

consistent with the results of the migratory area in the wound healing assays

(Revision Figure 10B & D). We concluded that both ELMO1 and IL22 could promote

the migration of MSC, but the effect of ELMO1 was much stronger than that of IL22.

A previous study demonstrated that IL22 promoted the migration of MSC ⁵⁷.

According to our further study above, we confirmed that IL22 accelerated MSC
migration. However, the effect of IL22 on MSC migration was weaker than that of
ELMO1.

Revision Figure 10

17. Fig 5: is this convincingly demonstrated to be an enthesis? Where is the H&E?

Response: Both human and SKG mouse tissue are shown in Figure 5. Regarding the

human tissue, the HE staining of the tissue is shown below (Revision Figure 11A),

and these results were not added in the manuscript because of the length limitation.

The HE staining of the SKG mice is shown in Figure 5E, and the enthesis results are

shown in Revision Figure 11B.

Revision Figure 11

18. TNF inhibits MSC differentiation by m⁶A effects on miRNA. but seen also in SLE
 and RA: so not specific for AS?

Response: N⁶-methyladenosine (m⁶A) is the most abundant internal mRNA
 modification and plays a critical role in the regulation of cell functions¹⁶. Recently,
 several studies have found that m⁶A modification affected MSC differentiation and
 senescence⁵⁸⁻⁶¹. Considering that MSC differentiation and senescence are related to
 the pathogenesis of other rheumatism diseases, including systemic lupus
 erythematosus (SLE) and rheumatoid arthritis (RA)⁶²⁻⁶⁴, it is possible that m⁶A
 modification may contribute to the MSC dysfunction of SLE or RA.

However, by scanning the literature concerning SLE or RA, m⁶A modification has
 been studied in PBMC, T cells and macrophages but not MSC⁶⁵⁻⁶⁷. In this study, for
 the first time, we demonstrated that the TNF- α /m⁶A modification/ELMO1 axis

triggers the directional migration of MSC in AS. This pathological mechanism has not
been reported in AS or other rheumatism diseases; therefore, we suggest that this axis
may be specific to AS. How m⁶A modification participates in the MSC dysfunctions
of SLE or RA still requires further studies in the future.

Responses to Reviewer 3

In this manuscript, Xie et al proposed ELMO1 as a novel potential therapeutic target
in ankylosing spondylitis (AS) and demonstrated the significance of m6A
modification in controlling the ELMO1 expression level in AS-MSC upon TNF- α
stimulation. Overall, the evidence is quite solid and the manuscript was well written,
with results being well organized and displayed. Nonetheless, some concerns need to
be addressed before the manuscript is suitable for publish. I have the following
suggestions to further strengthen this paper

We appreciate the positive comments by Reviewer 3 in the opening sentence. Based
on these helpful comments, we revised the manuscript as follows. The responses to
the comments by Reviewer 3 in this letter and the revised portions of the manuscript
are marked in green.

Major points:

1. The authors proposed that only high concentration of TNF- α (100 ng/mL) could
enhance the migration ability of AS-MSC. Does this concentration equal to that found
in AS patients? How do authors explain the phenomenon that HC-MSC and AS-MSC
displayed similar migration ability with TNF- α treatment at lower concentrations
(e.g., 10 and 50 ng/mL)?

Response: Thank you for your valuable comment. This issue was also mentioned by
Reviewer 2. Please see the response below.

In this study, we found that the migratory ability of HC-MSC and AS-MSC was
enhanced as the TNF- α concentration increased from 10 to 100 ng/ml. When treated

with 100 ng/ml TNF- α , AS-MSK showed a stronger migration ability than HC-MSK.
This difference between AS-MSK and HC-MSK was also observed in the 150 and 200
634 ng/ml TNF- α groups in our preliminary experiment. Actually, this level (>100 ng/ml)
is the pathological rather than physiological level of TNF- α *in vivo*. MSK constitute a
type of immunoregulatory cells *in vivo*⁴⁷, and TNF- α secreted by other immune cells
recruits MSK to specific sites to exert their functions^{48,49}. We suggest that two
functional forms (pathological and physiological) of MSK and TNF- α may exist in AS
patients according to our results.

① In pathological sites, such as entheses of AS, inflammatory cells, especially
macrophages, widely infiltrate and express a large amount of TNF- α at a pathological
level^{50,51}. The pathological level of TNF- α (>100 ng/ml) leads to the enhanced
migration capacity of AS-MSK, resulting in disease development as we suggested in
the discussion section (lines 353-360).

② In non-pathological lesion sites in AS patients, pathogenic factors, such as
infection or injury, induce normal inflammation to secrete physiological amounts of
TNF- α (much less than 100 ng/ml). This physiological level of TNF- α also recruits
moderate numbers of MSK to perform their function, and the migration ability of
these MSK is not abnormally enhanced.

Therefore, the different levels of TNF- α *in vivo* in AS patients may induce
physiological or pathological phenomena of MSK, and this result may also explain the
axial involvement (the site of higher TNF- α in AS patients, such as spine entheses) in
AS. We discussed this issue in the discussion section (lines 353-360). Actually, a

slight difference was observed in the 50 ng/ml group in the wound healing assays
without a statistical difference. As we addressed the comment by Reviewer 2, the
TNF- α stimulation decreased METTL14 expression in a
ubiquitin/proteasome-dependent manner (Revision Figure 3). Regarding the
mechanism, we suggest that this discrepancy between HC-MSC and AS-MSC may be
due to the different response capabilities of their ubiquitin/proteasome system when
treated with TNF- α at a higher level (such as 100 ng/ml). Certainly, this issue should
be addressed in future studies.

2. Did authors check the migratory capacity (as claimed) of HC-MSC or AS-MSC
when co-transplanted with macrophages into nude mice? This might mimic the real
TNF- α level in vivo.

Response: We appreciate this insightful comment. When we designed our *in vivo*
experiments, we considered co-transplanting MSC and macrophages into nude mice
as this comment suggests. However, several uncertainties may affect the results of
co-transplantation.

① The transplanted human macrophage may not secrete enough amount of TNF- α we
need in this experiment in nude mice.

② Transplanted human cells may result in immunoreaction in nude mice. To reduce
the effect of immunoreaction in the experiment, we chose to use one type of cells
(MSC) instead of two.

③ At the microscopic level, we observed MSC migration by detecting HLA⁺ cells by
an immunohistochemical assay. Co-transplanting two human cells may affect the

assessment of MSC migration.

Considering that TNF- α is the focus of our study, we chose to inject TNF- α rather
than macrophages into nude mice.

3. The authors used 3 siRNAs for testing and selected one with the highest
knockdown efficiency to construct shRNA for loss-of-function studies for both
ELMO1 and METTL14; however, this could not rule out the non-specific targeting.
One more shRNA for each gene is suggested to make the data more convincing.

Response: Thank you for this helpful comment. We selected siRNA3 of ELMO1 and
siRNA3 of METTL14 to construct the new shRNA, and migration assays were
performed.

As shown by the Western blot analysis, both shRNAs inhibited ELMO1 expression
(Revision Figure 12A). Additionally, the migration ability of MSC, as determined by
the Transwell migration assay and wound healing assay, was decreased after the
transfection with Lv-ELMO1-1 or Lv-ELMO1-2 (Revision Figure 12B & C).
Moreover, METTL14 expression in MSC was inhibited by both shRNA, and the
ELMO1 expression level was increased after the Lv-M14-1 or Lv-M14-2 transfection
(Revision Figure 12D). The migration ability of MSC was significantly increased
after both Lv-M14 transfections (Revision Figure 12E & F). These results indicate
that these siRNAs have a lower non-specific targeting effect on MSC, and the data in
the manuscript are convincing. Due to the limitation of the manuscript, these results
were not added in the manuscript. If the reviewer considers that these results should
be included, we will supplement them in the manuscript in the next revision.

Revision Figure 12

4. The authors proved the regulatory role of TNF- α in ELMO1 expression (Figure 3)

and proposed METTL14 as a key regulator in manipulating the m6A modification in

ELMO1 3' UTR (Figure 7,8 and 9), yet the connection between TNF- α and

METTL14 was neglected. To justify the feedback regulation between TNF- α and

AS-MSC, the mechanism on how TNF- α regulates METTL14 and how the

coordination affects AS-MSC migration require further exploration.

*Response: We appreciate this constructive comment regarding our study. According to*

*comment 5 by reviewer 1, which is consistent with this comment, we investigated the*

*mechanism by which TNF- α regulates the METTL14 level and suggest that TNF- α*

*affects METTL14 expression through a posttranslational modification pattern using*

*the ubiquitin-proteasome system. The response is attached below.*

*Previous studies have proven that protein degradation is one of the most common and*

*important posttranslational modification patterns 11. This posttranslational*

modification pattern occurs mainly through the autophagy-lysosome pathway and
ubiquitin-proteasome pathway 12,13. Therefore, we used MG132, which is a
proteasome inhibitor, and chloroquine (CQ), which is an autophagy inhibitor, to
investigate the role of these two pathways in TNF- α mediated METTL14 expression
in MSC. The results showed that the TNF- α treatment decreased the METTL14
protein level in MSC, and adding MG132 significantly antagonized this effect and
up-regulated METTL14 expression. Additionally, CQ did not affect METTL14
expression in the TNF- α treated MSC (Revision Figure 3B). These results indicate
that TNF- α decreased the METTL14 protein level in MSC through a posttranslational
modification pattern involving the ubiquitin-proteasome pathway.

Jian's study showed that TNF- α stimulation increased METTL14 expression at the
mRNA and protein levels in endothelial cells 14. This discrepancy may result from
the different cells investigated in the studies. However, how TNF- α affects METTL14
expression was not studied in their research. A previous study demonstrated that the
ubiquitin-proteasome system could be partially damaged in AS, contributing to the
development of inflammation ¹⁵. Combined with our supplemental results, we suggest
that the decrease in METTL14 induced by TNF- α through a posttranslational
modification pattern may result from a dysfunction in the ubiquitin-proteasome
system in AS-MSC after TNF- α treatment, which requires further investigation in the
future.

Revision Figure 3

5. The effect of m⁶A modification on mRNA transcripts is mediated by specific m⁶A
 readers. Did authors check whether the known m⁶A readers are responsible for
 ELMO1 mRNA degeneration after TNF- α stimulation?

Response: Thank you for this insightful comment.

m⁶A modification is regulated by writer enzymes (such as METTL14), eraser
 enzymes and reader enzymes. Specifically, m⁶A reader enzymes, including nuclear
 and cytoplasmic reader enzymes, regulate the target RNA expression through different
 mechanisms. YTHDC1, which is a major nuclear m⁶A reader enzyme, promotes exon
 splicing regulation. Regarding cytoplasmic m⁶A reader enzymes, YTHDF1 mediates
 the translation efficiency of target RNAs with specific m⁶A sites, and YTHDF2
 regulates m⁶A modified RNA decay and degeneration. Additionally, YTHDF3 affects
 the accessibility of YTHDF1 and YTHDF2 to target RNA. Moreover, YTHDC2
 exhibits different functions, such as affecting RNA structure, regulating translation
 efficiency, increasing RNA decay and binding protein complex⁶⁸. In our study, we
 demonstrated that METTL14 regulated ELMO1 expression through mRNA
 degeneration mechanism. Therefore, we suggest that YTHDF2, YTHDF3 or
 YTHDC2 may contribute to the METLL14-mediated ELMO1 expression in the

TNF- α treated MSC.

We conducted experiments to determine which m⁶A reader enzyme contributed to

METTL14-mediated ELMO1 mRNA degeneration. ① CLIP assays using antibodies

against YTHDC1-2 and YTHDF1-3 were performed to detect ELMO1 mRNA

binding the m⁶A reader enzyme. The results showed that the %Input levels in the

YTHDC1-2 and YTHDF1 groups were equal to those in the IgG group, and

the %Input levels in the YTHDF2-3 groups were higher than those in the IgG group

(Revision Figure 13A). These results indicate that YTHDF2 and 3 could bind ELMO1

mRNA. ② We further designed three siRNAs for YTHDF2 and YTHDF3, and the

siRNA with the highest inhibitory efficiency was chosen for the RNA stability assays.

After inhibiting YTHDF2 or YTHDF3 in MSC, the stability of ELMO1 mRNA was

increased, and its degeneration rate was significantly decreased (Revision Figure 13B).

These results demonstrate that YTHDF2 and YTHDF3 were responsible for the

ELMO1 mRNA degeneration in the TNF- α treated MSC. These data were added in

the manuscript as Supplemental Figure 7. Please see lines 317-321.

**Revision Figure 13**

6. Please provide the Co-IP results as mentioned in line 200 (supplementary figure 3).

Response: We apologize for our carelessness. In the first submission, we uploaded the

Co-IP results in Supplemental Tables 5 (ELMO1 group) and 6 (IgG group) as two
Excel files. This style of file may have failed to display properly for the reviewers. We
changed these two files to Word files. Please see the new Supplemental Tables 5 and
6.

7. Can authors provide results of METTL14 expression at the protein level in
METTL14 knockdown or overexpression experiments (Supplementary 4)?

Response: Thank you for your comment. METTL14 expression at the protein level in
the METTL14 knockdown and overexpression experiments is shown in Figure 7A
(knockdown) and Figure 8A (overexpression).

8. The assay using nude mice for transplanting MSC seems to reflect cell proliferation
rather than the directional migration of transplanted cells. Could the authors explain a
bit more?

Response: We appreciated this helpful comment.

Minor comment 2 by reviewer 1 also concerned the effect of MSC proliferation on the
results of the *in vivo* migration assay. As shown in Revision Figure 6, inhibiting or
overexpressing ELMO1 did not affect the proliferation ability of MSC. According to
this comment, we detected the proliferation ability of MSC after Lv-METTL14 or
OE-METTL14 transfection. Consistent with the results of ELMO1, Lv-METTL14
and OE-METTL14 did not affect MSC proliferation (Revision Figure 14), indicating
that the *in vivo* migration assay reflected the real migration ability rather proliferative
activity. These results were added in Supplemental Figure 6. This assay was also used
to determine the cell migration ability in a previous study⁶⁹.

Revision Figure 14

9. The figure legends should provide description of related experiments rather than
 discussion of the results. More importantly, a legend should be understandable
 without the need to consult the text. It is strongly encouraged that the authors add
 necessary details in figure legends. For example, what do different colors in figure 3H
 stand for? What do specific abbreviations indicate, such as “W/O, TNFi”? In addition,
 please add the description of error bars in the corresponding figure legends.

**Response:** We apologize for our unclear presentation in the figure legends. We revised
 our descriptions and added details in the legends. Please see the legends section
 (963-1154).

Minor points:

1. The introduction part (Lines 122-129) just simply repeated the sentences in the
 Abstract (lines 66-73). Please revise to avoid that.

**Response:** Thank you for your comment. We revised the last paragraph of the
 introduction section. Please see lines 123-131.

2. The sub title in Page 11 is inappropriate. METTL14 will not lead to low m6A
 modification, please correct.

Response: The subtitle was changed to “Decreasing METTL14 expression leads to
lower m6A modification and RNA degradation of ELMO1 in TNF- α -treated
AS-MSK”. Please see lines 253-254.

3. Line 236, “we constructed Av-ELMO1 for SKG mouse treatment.” Can authors
provide experimental evidence that the expression level of ELMO1 in SKG mice
treated with Av-ELMO1 is indeed reduced?

Response: As we explained in the response to comment 4 by Reviewer 1, we isolated
MSC from SKG mice at 0, 4 and 8 week after curdlan induction and then detected
their ELMO1 expression. The results showed that ELMO1 expression in the SKG
mice MSC increased from 0 to 8 weeks after the curdlan induction in the PBS group
and Av-NC group. ELMO1 expression in the SKG mice MSC was significantly
inhibited in the Av-ELMO1 group at 4 and 8 weeks compared to that in the other two
groups (Revision Figure 2A). Additionally, the results of immunofluorescence assays
showed that ELMO1 expression in MSC increased after the induction at 8 weeks and
that ELMO1 expression in MSC was lower than that in the other two groups at 8
826 weeks (Revision Figure 2B). These results show that ELMO1 expression was induced
in the SKG mice after the curdlan induction and that Av-ELMO1 could target MSC
and inhibit ELMO1 expression.

Revision Figure 2

4. Line 296, the statement perhaps should be rephrased to “CLIP-PCR results showed
that ELMO1 was enriched in the METTL14 group than in the IgG group.”

**Response:** This sentence was rephrased. Please see lines 305-306

5. It’s necessary to add scale bars in corresponding figure, including but not limited to
figure 5E, 5G.

**Response:** Thank you very much for this comment. The scale bars were added.

6. Please annotate the sample name in Figure 5G.

**Response:** The local tissue in the SKG mice shown in Figure 5G is ankle entheses.

**Response:** Please see lines 249-252.

7. Figure 6, panels F to I could be moved to supplementary figures.

**Response:** Panels F to I in Figure 6 were moved to the new Supplemental Figure 5.

**Response:** The related descriptions in the Results and Figure legends were revised.

8. Figure 9, panels D to F should be merged into one.

**Response:** Panels D, E and F in Figure 9 were merged into panel D. Please see the
new Figure 9.

9. When citing references, it is highly recommended to cite original research articles
rather than review articles.

Response: Thank you for this helpful comment. The references were checked, and
several improper citations were removed.

Responses to Reviewer 4

The manuscript by Xie et al. entitled “TNF-a-mediated m6A modification of ELMO1
triggers directional migration of mesenchymal stem cell in ankylosing spondylitis”
aims at investigating the role of m6A modification of ELMO1 by TNF-a in migration
of MSC. TNFa leads to increase migration of MSC through increase of ELMO1
expression. Silencing of ELMO1 leads to reduction of MSC migration as well as
arthritis symptoms and overexpression of ELMO1 leads to increase of HC MSC
migration. m6A modification of ELMO1 is regulated by METTL14 and it is
decreased in high concentration of TNFa-treated MSC.

This is an interesting and timely manuscript. The experiments have been carefully
carried out and most of the conclusions drawn are justified. There are no ethical
concerns arising. The authors' claims are mostly convincing. On that account, I have
only some minor suggestions below.

We appreciate the positive comments by Reviewer 4 in the opening sentence. Based
on these helpful comments, we revised the manuscript as follows. The responses to
the comments by Reviewer 4 in this letter and the revised portions of the manuscript
are marked in orange.

1. When comparing more than three groups, usually we use one way ANOVA or
Kruskal Wallis test. However the statistical information is lacking in method section.
The number of samples are relatively low (N=6 to 9), I think the authors should use
Kruskal Wallis test. In addition, the significance should be presented by comparing to
each group (figure 4,5,7,8,9). As presented in method section of statistics, why the

author used mean with SD? I think using nonparametric statistical method is suitable
for current study, because number of each group was small.

Response: Thank you for this constructive comment. We apologize for our
carelessness and not indicating the statistical method in our manuscript. We provided
the details of our statistical methods in lines 746-751.

In this study, the in vitro data presented were derived from at least three independent
experiments using at least triplicate samples, except for the six HC-MSK and six
AS-MSK used in the RNA-seq assays. For the two-group comparisons, a 2-tailed
Student's t-test was used, such as in Figure 1D and Figure 2C. For the three or more
different group comparisons, a one-way ANOVA followed by Bonferroni's post hoc
comparisons was used, such as in Figure 4 and Figure 7-8.

The sample number (n) was nine in a large proportion of the in vitro experiments in
this study. This sample number may not be very large, but all data with a
Shapiro–Wilk test $P > 0.05$ were considered to fit a normal distribution. Therefore, we
used parametric statistical methods rather than nonparametric statistical methods
because nonparametric statistical methods may lead to a lower power of the tests.

2. The results are commented in the most of figure legends, whereas there should be
no comment of results in figure legends.

Response: We appreciate this helpful comment. We revised our descriptions and
added details in the legends. Please see lines 963-1153.

3. In figure 2B, 4C, H, 7H, and 8H, injected MSC migrated less or more into TNFa
injected site. In case of less migrated mice, it seems that MSC are staying at first

injected site. Is there any data available that supports the authors' conclusions?

Response: In the in vivo migration assays, we used the bioluminescence area to
measure MSC migration. The longer the distance the MSC migrated, the larger the
area shown in the bioluminescence analysis assay. In our study, we suggest that all
MSC could migrate after the injection and that TNF- α simply promoted their
migration ability. Therefore, in the less migrated mice, MSC could also gradually
migrate rather than stay at the injected site without movement. However, the
bioluminescence area of the less migrated mice could be much less than that of the
other mice.

4. In figure 5, authors showed that silencing of ELMO1 has protective effect in
arthritis. However, in figure 5 legend (line 954-957), the explanation is opposite.
Please revise the sentence.

Response: We apologize for our carelessness in the presentation. The two sentences
were revised. Please see lines 1031-1034.

5. In figure 5, Did the arthritis score and incidence were significantly differ between
AV-ELMO1 group vs PBS or AV-NC. The figure lacks the significance. Furthermore,
the MSC is the progenitor of osteoblast, and inhibiting MSC in SKG mice could
ameliorate syndesmophyte formation in vivo SpA model. Did the author tried to
quantify the syndesmophyte formation in micro CT or histology?

Response: Thank you for this helpful comment. The differences in the arthritis score
among the PBS group, Av-NC group and Av-ELMO1 group were analyzed, and the
score of the Av-ELMO1 group was significantly lower than that of the PBS group or

Av-NC group. Regarding the incidence rate, the disease developed in all SKG mice in
the three groups at week 7 after induction, and statistical differences were observed at
936 weeks 3, 4 and 5. The marker of the statistic difference was added. Please see Figure
5.

In addition, in this study, we did not attempt to quantify the syndesmophyte formation
because of two reasons. On the one hand, at the histological level, new bone
formation is always accompanied by bone resorption (as shown in Figure 5E).
Therefore, it is difficult to simply mark the area of new bone formation, and it may be
inaccurate to only mark the area of new bone formation to represent the whole
pathological tissue. On the other hand, syndesmophyte formation resulting from
enthesitis inflammation occurs on the surface of the vertebrae, which leads to a higher
bone volume parameter in micro-CT. However, the bone mass inside the vertebrae is
always reduced because of the joint fusion stiffness mediated disuse osteoporosis,
which leads to a lower bone parameter in micro-CT. Although micro-CT
quantification is applied to measure the bone volume of the osteoporosis mouse model,
we suggest that quantifying syndesmophyte formation may not reflect the authenticity
very well based on these two reasons. Therefore, we only analyzed syndesmophyte
formation qualitatively in our manuscript.

6. Furthermore, the incidence rate reach similar between PBS, Av-NC, and
Av-ELMO1 group (figure 5C). Do authors has hypothesis why the effect of ELMO1
inhibition disappeared (or weakened) after 7 weeks ?

Response: In our study, we treated the SKG mice with PBS, Av-NC or Av-ELMO1

once at week 0. The disease incidence rate in the Av-ELMO1 group was delayed and
was lower than that in the PBS and Av-NC groups at 2 to 4 weeks, indicating the
therapeutic effect of Av-ELMO1 in SKG mice. At weeks 7 and 8, the incidence rate
was similar among the three groups (arthritis score > 0 is considered disease
incidence), but the arthritis score of the Av-ELMO1 group was still significantly
lower than that of the other two groups. We suggest that this phenomenon resulted
from the treatment times of Av-ELMO1, and the SKG mice in the Av-ELMO1 group
began to exhibit symptoms after the effect of Av-ELMO1 disappeared at weeks 7 to 8.
If the reviewer considers that the disease incidence rate does not accurately reflect the
effect of Av-ELMO1, we could remove this panel from the results in the next revision.

**References**

- 1. Schaker, K. et al. The bipartite rac1 Guanine nucleotide exchange factor
engulfment and cell motility 1/dedicator of cytokinesis 180 (elmo1/dock180)
protects endothelial cells from apoptosis in blood vessel development. *J. Biol.*
*Chem.* **290**, 6408-6418 (2015).
- 2. Arandjelovic, S. et al. A noncanonical role for the engulfment gene ELMO1 in
neutrophils that promotes inflammatory arthritis. *Nat. Immunol.* **20**, 141-151
(2019).
- 3. Zhang, G. et al. A novel interaction between the SH2 domain of signaling adaptor
protein Nck-1 and the upstream regulator of the Rho family GTPase Rac1
engulfment and cell motility 1 (ELMO1) promotes Rac1 activation and cell
motility. *J. Biol. Chem.* **289**, 23112-23122 (2014).
- 4. Lu, M. et al. A Steric-inhibition model for regulation of nucleotide exchange via
the Dock180 family of GEFs. *Curr. Biol.* **15**, 371-377 (2005).
- 5. Lau, M. C. et al. Genetic association of ankylosing spondylitis with TBX21
influences T-bet and pro-inflammatory cytokine expression in humans and SKG
mice as a model of spondyloarthritis. *Ann. Rheum. Dis.* **76**, 261-269 (2017).
- 6. Jeong, H. et al. Estrogen attenuates the spondyloarthritis manifestations of the
SKG arthritis model. *Arthritis Res. Ther.* **19**, 198 (2017).
- 7. Benham, H. et al. Interleukin-23 mediates the intestinal response to microbial
beta-1,3-glucan and the development of spondyloarthritis pathology in SKG
mice. *Arthritis Rheumatol.* **66**, 1755-1767 (2014).

- 8. Xu, X., Su, Y., Wu, K., Pan, F. & Wang, A. DOCK2 contributes to
endotoxemia-induced acute lung injury in mice by activating proinflammatory
macrophages. *Biochem. Pharmacol.*, 114399 (2020).
- 9. Watanabe, M. et al. DOCK2 and DOCK5 act additively in neutrophils to regulate
chemotaxis, superoxide production, and extracellular trap formation. *J.*
*Immunol.* **193**, 5660-5667 (2014).
- 10. Morikawa, M., Tanaka, Y., Cho, H. S., Yoshihara, M. & Hirokawa, N. The
Molecular Motor KIF21B Mediates Synaptic Plasticity and Fear Extinction by
Terminating Rac1 Activation. *Cell Rep.* **23**, 3864-3877 (2018).
- 11. Goldberg, A. L. Protein degradation and protection against misfolded or damaged
proteins. *Nature.* **426**, 895-899 (2003).
- 12. Klionsky, D. J. & Emr, S. D. Autophagy as a regulated pathway of cellular
degradation. *Science.* **290**, 1717-1721 (2000).
- 13. Lecker, S. H., Goldberg, A. L. & Mitch, W. E. Protein degradation by the
ubiquitin-proteasome pathway in normal and disease states. *J. Am. Soc. Nephrol.*
**17**, 1807-1819 (2006).
- 14. Jian, D. et al. METTL14 aggravates endothelial inflammation and atherosclerosis
by increasing FOXO1 N6-methyladenosine modifications. *Theranostics.* **10**,
8939-8956 (2020).
- 15. Wright, C. et al. Ankylosing spondylitis monocytes show upregulation of proteins
involved in inflammation and the ubiquitin proteasome pathway. *Ann. Rheum.*
*Dis.* **68**, 1626-1632 (2009).

- 16. Lee, M., Kim, B. & Kim, V. N. Emerging roles of RNA modification: m(6)A and
U-tail. *Cell*. **158**, 980-987 (2014).
- 17. Uccelli, A., Moretta, L. & Pistoia, V. Mesenchymal stem cells in health and
disease. *Nat. Rev. Immunol.* **8**, 726-736 (2008).
- 18. Kfoury, Y. & Scadden, D. T. Mesenchymal cell contributions to the stem cell
niche. *Cell Stem Cell*. **16**, 239-253 (2015).
- 19. Abbuehl, J. P., Tatarova, Z., Held, W. & Huelsken, J. Long-Term Engraftment of
Primary Bone Marrow Stromal Cells Repairs Niche Damage and Improves
Hematopoietic Stem Cell Transplantation. *Cell Stem Cell*. **21**, 241-255 (2017).
- 20. Pinho, S. et al. PDGFRalpha and CD51 mark human nestin+ sphere-forming
mesenchymal stem cells capable of hematopoietic progenitor cell expansion. *J.*
*Exp. Med.* **210**, 1351-1367 (2013).
- 21. Luz-Crawford, P. et al. Mesenchymal stem cell repression of Th17 cells is
triggered by mitochondrial transfer. *Stem Cell Res. Ther.* **10**, 232 (2019).
- 22. Vellasamy, S. et al. Human mesenchymal stromal cells modulate T-cell immune
response via transcriptomic regulation. *Cytotherapy*. **18**, 1270-1283 (2016).
- 23. Luz-Crawford, P. et al. Mesenchymal Stem Cell-Derived Interleukin 1 Receptor
Antagonist Promotes Macrophage Polarization and Inhibits B Cell Differentiation.
*Stem Cells*. **34**, 483-492 (2016).
- 24. Khare, D. et al. Mesenchymal Stromal Cell-Derived Exosomes Affect mRNA
Expression and Function of B-Lymphocytes. *Front Immunol.* **9**, 3053 (2018).
- 25. Lo, S. C. et al. Mesenchymal Stem Cell-Derived Extracellular Vesicles as

- Mediators of Anti-Inflammatory Effects: Endorsement of Macrophage
Polarization. *Stem Cells Transl Med.* **6**, 1018-1028 (2017).
- 26. Qiu, X. et al. Mesenchymal stem cells and extracellular matrix scaffold promote
muscle regeneration by synergistically regulating macrophage polarization
toward the M2 phenotype. *Stem Cell Res. Ther.* **9**, 88 (2018).
- 27. Samsonraj, R. M. et al. Concise Review: Multifaceted Characterization of Human
Mesenchymal Stem Cells for Use in Regenerative Medicine. *Stem Cells Transl
Med.* **6**, 2173-2185 (2017).
- 28. Bianco, P. "Mesenchymal" stem cells. *Annu Rev Cell Dev Biol.* **30**, 677-704
(2014).
- 29. Ambrosi, T. H., Longaker, M. T. & Chan, C. A Revised Perspective of Skeletal
Stem Cell Biology. *Front Cell Dev Biol.* **7**, 189 (2019).
- 30. Kurenkova, A. D., Medvedeva, E. V., Newton, P. T. & Chagin, A. S. Niches for
Skeletal Stem Cells of Mesenchymal Origin. *Front Cell Dev Biol.* **8**, 592 (2020).
- 31. Zhou, B. O., Yue, R., Murphy, M. M., Peyer, J. G. & Morrison, S. J.
Leptin-receptor-expressing mesenchymal stromal cells represent the main source
of bone formed by adult bone marrow. *Cell Stem Cell.* **15**, 154-168 (2014).
- 32. Park, D. et al. Endogenous bone marrow MSCs are dynamic, fate-restricted
participants in bone maintenance and regeneration. *Cell Stem Cell.* **10**, 259-272
(2012).
- 33. Zhao, H. et al. The suture provides a niche for mesenchymal stem cells of
craniofacial bones. *Nat. Cell Biol.* **17**, 386-396 (2015).

- 34. Shi, Y. et al. Gli1 identifies osteogenic progenitors for bone formation and
fracture repair. *Nat. Commun.* **8**, 2043 (2017).
- 35. Pineault, K. M., Song, J. Y., Kozloff, K. M., Lucas, D. & Wellik, D. M. Hox11
expressing regional skeletal stem cells are progenitors for osteoblasts,
chondrocytes and adipocytes throughout life. *Nat. Commun.* **10**, 3168 (2019).
- 36. Wang, J. et al. Circulating levels of Th1 and Th2 chemokines in patients with
ankylosing spondylitis. *Cytokine*. **81**, 10-14 (2016).
- 37. Sezer, U. et al. Serum cytokine levels and periodontal parameters in ankylosing
spondylitis. *J. Periodontal Res.* **47**, 396-401 (2012).
- 38. Deveci, H., Turk, A. C., Ozmen, Z. C., Demir, A. K. & Say, C. S. Biological and
genetic evaluation of IL-23/IL-17 pathway in ankylosing spondylitis patients.
*Cent. Eur. J. Immunol.* **44**, 433-439 (2019).
- 39. Croft, M. & Siegel, R. M. Beyond TNF: TNF superfamily cytokines as targets for
the treatment of rheumatic diseases. *Nat. Rev. Rheumatol.* **13**, 217-233 (2017).
- 40. Xie, J., Wang, Z. & Wang, W. Semaphorin 4D Induces an Imbalance of
Th17/Treg Cells by Activating the Aryl Hydrocarbon Receptor in Ankylosing
Spondylitis. *Front Immunol.* **11**, 2151 (2020).
- 41. Akhtari, M. et al. Activation of adenosine A2A receptor induced interleukin-23
mRNA expression in macrophages of ankylosing spondylitis patients. *Cytokine*.
**128**, 154997 (2020).
- 42. Xie, Z. et al. Imbalance Between Bone Morphogenetic Protein 2 and Noggin
Induces Abnormal Osteogenic Differentiation of Mesenchymal Stem Cells in

- Ankylosing Spondylitis. *Arthritis Rheumatol.* **68**, 430-440 (2016).
- 43. Liu, C. H. et al. HLA-B27-mediated activation of TNAP phosphatase promotes
pathogenic syndesmophyte formation in ankylosing spondylitis. *J. Clin. Invest.*
**129**, 5357-5373 (2019).
- 44. Gravallese, E. M. & Schett, G. Effects of the IL-23-IL-17 pathway on bone in
spondyloarthritis. *Nat. Rev. Rheumatol.* **14**, 631-640 (2018).
- 45. Sieper, J., Poddubnyy, D. & Miossec, P. The IL-23-IL-17 pathway as a
therapeutic target in axial spondyloarthritis. *Nat. Rev. Rheumatol.* **15**, 747-757
(2019).
- 46. Pedersen, S. J. & Maksymowych, W. P. The Pathogenesis of Ankylosing
Spondylitis: an Update. *Curr. Rheumatol. Rep.* **21**, 58 (2019).
- 47. Bernardo, M. E. & Fibbe, W. E. Mesenchymal stromal cells: sensors and
switchers of inflammation. *Cell Stem Cell.* **13**, 392-402 (2013).
- 48. Wang, Y. et al. TNF-alpha-induced LRG1 promotes angiogenesis and
mesenchymal stem cell migration in the subchondral bone during osteoarthritis.
*Cell Death Dis.* **8**, e2715 (2017).
- 49. Xiao, Q. et al. TNF-alpha increases bone marrow mesenchymal stem cell
migration to ischemic tissues. *Cell Biochem. Biophys.* **62**, 409-414 (2012).
- 50. Braun, J. et al. Use of immunohistologic and in situ hybridization techniques in
the examination of sacroiliac joint biopsy specimens from patients with
ankylosing spondylitis. *Arthritis Rheum.* **38**, 499-505 (1995).
- 51. Francois, R. J., Neure, L., Sieper, J. & Braun, J. Immunohistological examination

of open sacroiliac biopsies of patients with ankylosing spondylitis: detection of
tumour necrosis factor alpha in two patients with early disease and transforming
growth factor beta in three more advanced cases. *Ann. Rheum. Dis.* **65**, 713-720
(2006).

52. Schett, G. et al. Enthesitis: from pathophysiology to treatment. *Nat. Rev.*
*Rheumatol.* **13**, 731-741 (2017).

53. Benjamin, M. et al. Microdamage and altered vascularity at the enthesis-bone
interface provides an anatomic explanation for bone involvement in the
HLA-B27-associated spondylarthritides and allied disorders. *Arthritis Rheum.* **56**,
224-233 (2007).

54. Berthelot, J. M., Le Goff, B. & Maugars, Y. Bone marrow mesenchymal stem
cells in rheumatoid arthritis, spondyloarthritis, and ankylosing spondylitis:
problems rather than solutions? *Arthritis Res. Ther.* **21**, 239 (2019).

55. Dabbah, M. et al. Multiple myeloma cells promote migration of bone marrow
mesenchymal stem cells by altering their translation initiation. *J Leukoc Biol.* **100**,
761-770 (2016).

56. Geng, L. et al. Association of TNF-alpha with impaired migration capacity of
mesenchymal stem cells in patients with systemic lupus erythematosus. *J*
*Immunol Res.* **2014**, 169082 (2014).

57. El-Zayadi, A. A. et al. Interleukin-22 drives the proliferation, migration and
osteogenic differentiation of mesenchymal stem cells: a novel cytokine that could
contribute to new bone formation in spondyloarthropathies. *Rheumatology*

- (Oxford). **56**, 488-493 (2017).
- 58. Wu, Y. et al. Mettl3-mediated m(6)A RNA methylation regulates the fate of bone
marrow mesenchymal stem cells and osteoporosis. *Nat. Commun.* **9**, 4772 (2018).
- 59. Tian, C., Huang, Y., Li, Q., Feng, Z. & Xu, Q. Mettl3 Regulates Osteogenic
Differentiation and Alternative Splicing of Vegfa in Bone Marrow
Mesenchymal Stem Cells. *Int. J. Mol. Sci.* **20**, (2019).
- 60. Wu, Z. et al. METTL3 counteracts premature aging via m6A-dependent
stabilization of MIS12 mRNA. *Nucleic Acids Res.* **48**, 11083-11096 (2020).
- 61. Yan, G. et al. m(6)A Methylation of Precursor-miR-320/RUNX2 Controls
Osteogenic Potential of Bone Marrow-Derived Mesenchymal Stem Cells. *Mol*
*Ther Nucleic Acids.* **19**, 421-436 (2020).
- 62. Gao, L. et al. Bone Marrow-Derived Mesenchymal Stem Cells From Patients
With Systemic Lupus Erythematosus Have a Senescence-Associated Secretory
Phenotype Mediated by a Mitochondrial Antiviral Signaling
Protein-Interferon-beta Feedback Loop. *Arthritis Rheumatol.* **69**, 1623-1635
(2017).
- 63. Tang, Y. et al. Activated NF-kappaB in bone marrow mesenchymal stem cells
from systemic lupus erythematosus patients inhibits osteogenic differentiation
through downregulating Smad signaling. *Stem Cells Dev.* **22**, 668-678 (2013).
- 64. Lamas, J. R. et al. RNA sequencing of mesenchymal stem cells reveals a blocking
of differentiation and immunomodulatory activities under inflammatory
conditions in rheumatoid arthritis patients. *Arthritis Res. Ther.* **21**, 112 (2019).

- 65. Luo, Q. et al. The study of METTL14, ALKBH5, and YTHDF2 in peripheral
blood mononuclear cells from systemic lupus erythematosus. *Mol Genet Genomic*
*Med.* **8**, e1298 (2020).
- 66. Guo, G. et al. Disease Activity-Associated Alteration of mRNA m(5) C
Methylation in CD4(+) T Cells of Systemic Lupus Erythematosus. *Front Cell*
*Dev Biol.* **8**, 430 (2020).
- 67. Wang, J., Yan, S., Lu, H., Wang, S. & Xu, D. METTL3 Attenuates LPS-Induced
Inflammatory Response in Macrophages via NF-kappaB Signaling Pathway.
*Mediators Inflamm.* **2019**, 3120391 (2019).
- 68. Yang, Y., Hsu, P. J., Chen, Y. S. & Yang, Y. G. Dynamic transcriptomic m(6)A
decoration: writers, erasers, readers and functions in RNA metabolism. *Cell Res.*
**28**, 616-624 (2018).
- 69. Zou, Y. et al. Long noncoding RNA LERFS negatively regulates rheumatoid
synovial aggression and proliferation. *J. Clin. Invest.* **128**, 4510-4524 (2018).

Reviewer comments, second round –

Reviewer #1 (Remarks to the Author):

The authors have significantly improved their manuscript and provide thoughtful answers to previous criticisms. The paper is more focused without the filopodia claims. Also, the paper is significantly improved with the in vivo data that Curdulan upregulates Elmo1 expression. The mechanisms of MTL4 stability and the RNA-seq data adds to the paper and reveal the complexity of the system. While pleiotropic effects are observed, Elmo1 is clearly a target. I think these studies add to the manuscript and conclusions.

While the novelty on Elmo in migration remains a limitation of the study, this is now well balanced with the added new experiments and the global conclusions.

Minor comment #1. I still find that it is irrelevant to show co-ip of Elmo1 with Dock1, Dock4 and Dock5, but I leave it up to the authors. The new Rac pull-down data is very convincing - congrats to the authors.

Minor comment #2. The authors should introduce the Chang L. 2020 (Nat Comms) when discussing the mechanisms of Elmo-Dock interaction and activation of the complex for Rac activation.

Jean-Francois Cote

Reviewer #2 (Remarks to the Author):

The authors have adequately addressed my concerns and queries.

Reviewer #3 (Remarks to the Author):

The authors have fully addressed my questions.

Reviewer #4 (Remarks to the Author):

I have no remarks for the revised version of the manuscript by Xie et al " TNF- α -mediated m6A modification of ELMO1 triggers directional migration of mesenchymal stem cell in ankylosing spondylitis". The authors have properly and fully addressed all the questions raised during the first revision.

Response To Reviewers Letter

Dear Reviewers:

Thank you for the constructive comments concerning our manuscript entitled “TNF- α -mediated m⁶A modification of ELMO1 triggers directional migration of mesenchymal stem cell in ankylosing spondylitis” (Manuscript ID: NCOMMS-20-35339A). These comments were valuable and very helpful for improving our manuscript to better demonstrate the important significance of our research. We have carefully reviewed all comments and completed point-by-point revisions. The responses to the comments in this letter and the revised portions of the manuscript are marked in red. We appreciate the work of Reviewers and hope that the revisions will meet with approval. We will be glad to respond to any further comments that you may have.

Yours sincerely,

Huiyong Shen

Department of Orthopedics, The Eighth Affiliated Hospital, Sun Yat-sen University,
3025# Shennan Road, Shenzhen, 518000, P.R. China.

Tel: +86 139 2227 6368

Fax: +86 20 8133 2612

Email: shenhuiy@mail.sysu.edu.cn

Responses to Reviewer 1

The authors have significantly improved their manuscript and provide thoughtful
answers to previous criticisms. The paper is more focused without the filiopodia
claims. Also, the paper is significantly improved with the in vivo data that Curdulan
upregulates Elmo1 expression. The mechanisms of MTTL4 stability and the RNA-seq
data adds to the paper and reveal the complexity of the system. While pleotropic
effects are observed, Elmo1 is clearly a target. I think these studies add to the
manuscript and conclusions.

While the novelty on Elmo in migration remains a limitation of the study, this is now
well balanced with the added new experiments and the global conclusions.

Minor comment #1. I still find that it is irrelevant to show co-ip of Elmo1 with Dock1,
Dock4 and Dock5, but I leave it up to the authors. The new Rac pull-down data is
very convincing - congrats to the authors.

**Response: Thank you very much for the comment. Due to the results of Co-IP and**
**LC-MS/MS, we chose to retain the data of DOCK1, 4 and 5 in our manuscript.**

Minor comment #2. The authors should introduce the Chang L. 2020 (Nat Comms)
when discussing the mechanisms of Elmo-Dock interaction and activation of the
complex for Rac activation.

**Response: Thank you very much for the comment. The mention reference has been**
**added in the Discussion part.**

**Reviewer #2 (Remarks to the Author):**

The authors have adequately addressed my concerns and queries.

**Reviewer #3 (Remarks to the Author):**

The authors have fully addressed my questions.

**Reviewer #4 (Remarks to the Author):**

I have no remarks for the revised version of the manuscript by Xie et al " TNF- α
-mediated m6A modification of ELMO1 triggers directional migration of
mesenchymal stem cell in ankylosing spondylitis". The authors have properly and
fully addressed all the questions raised during the first revision.

**No response needed.**

Response To Reviewers Letter

Dear Reviewers:

Thank you for the constructive comments concerning our manuscript entitled “TNF- α -mediated m⁶A modification of ELMO1 triggers directional migration of mesenchymal stem cell in ankylosing spondylitis” (Manuscript ID: NCOMMS-20-35339A). These comments were valuable and very helpful for improving our manuscript to better demonstrate the important significance of our research. We have carefully reviewed all comments and completed point-by-point revisions. The responses to the comments in this letter and the revised portions of the manuscript are marked in red. We appreciate the work of Reviewers and hope that the revisions will meet with approval. We will be glad to respond to any further comments that you may have.

Yours sincerely,

Huiyong Shen

Department of Orthopedics, The Eighth Affiliated Hospital, Sun Yat-sen University,
3025# Shennan Road, Shenzhen, 518000, P.R. China.

Tel: +86 139 2227 6368

Fax: +86 20 8133 2612

Email: shenhuiy@mail.sysu.edu.cn

Responses to Reviewer 1

The authors have significantly improved their manuscript and provide thoughtful
answers to previous criticisms. The paper is more focused without the filiopodia
claims. Also, the paper is significantly improved with the in vivo data that Curdulan
upregulates Elmo1 expression. The mechanisms of MTTL4 stability and the RNA-seq
data adds to the paper and reveal the complexity of the system. While pleiotropic
effects are observed, Elmo1 is clearly a target. I think these studies add to the
manuscript and conclusions.

While the novelty on Elmo in migration remains a limitation of the study, this is now
well balanced with the added new experiments and the global conclusions.

Minor comment #1. I still find that it is irrelevant to show co-ip of Elmo1 with Dock1,
Dock4 and Dock5, but I leave it up to the authors. The new Rac pull-down data is
very convincing - congrats to the authors.

**Response: Thank you very much for the comment. Due to the results of Co-IP and**
**LC-MS/MS, we chose to retain the data of DOCK1, 4 and 5 in our manuscript.**

Minor comment #2. The authors should introduce the Chang L. 2020 (Nat Comms)
when discussing the mechanisms of Elmo-Dock interaction and activation of the
complex for Rac activation.

**Response: Thank you very much for the comment. The mention reference has been**
**added in the Discussion part.**

**Reviewer #2 (Remarks to the Author):**

The authors have adequately addressed my concerns and queries.

**Reviewer #3 (Remarks to the Author):**

The authors have fully addressed my questions.

**Reviewer #4 (Remarks to the Author):**

I have no remarks for the revised version of the manuscript by Xie et al " TNF- α
-mediated m6A modification of ELMO1 triggers directional migration of
mesenchymal stem cell in ankylosing spondylitis". The authors have properly and
fully addressed all the questions raised during the first revision.

**No response needed.**